# Computational Complexity of Learning Neural Networks: Smoothness and Degeneracy

**Amit Daniely**
Hebrew University and Google
amit.daniely@mail.huji.ac.il

**Nathan Srebro**
TTI-Chicago
nati@ttic.edu

**Gal Vardi**
TTI-Chicago and Hebrew University
galvardi@ttic.edu

## Abstract

Understanding when neural networks can be learned efficiently is a fundamental question in learning theory. Existing hardness results suggest that assumptions on both the input distribution and the network's weights are necessary for obtaining efficient algorithms. Moreover, it was previously shown that depth-2 networks can be efficiently learned under the assumptions that the input distribution is Gaussian, and the weight matrix is non-degenerate. In this work, we study whether such assumptions may suffice for learning deeper networks and prove negative results. We show that learning depth-3 ReLU networks under the Gaussian input distribution is hard even in the smoothed-analysis framework, where a random noise is added to the network's parameters. It implies that learning depth-3 ReLU networks under the Gaussian distribution is hard even if the weight matrices are non-degenerate. Moreover, we consider depth-2 networks, and show hardness of learning in the smoothed-analysis framework, where both the network parameters and the input distribution are smoothed. Our hardness results are under a well-studied assumption on the existence of local pseudorandom generators.

## 1 Introduction

The computational complexity of learning neural networks has been extensively studied in recent years, and there has been much effort in obtaining both upper bounds and hardness results. Nevertheless, it is still unclear when neural networks can be learned in polynomial time, namely, under what assumptions provably efficient algorithms exist.

Existing results imply hardness already for learning depth-2 ReLU networks in the standard PAC learning framework (e.g., [31, 14]). Thus, without any assumptions on the input distribution or the network's weights, efficient learning algorithms might not be achievable. Even when assuming that the input distribution is Gaussian, strong hardness results were obtained for depth-3 ReLU networks [16, 11], suggesting that assumptions merely on the input distribution might not suffice. Also, a hardness result by Daniely and Vardi [15] shows that strong assumptions merely on the network's weights (without restricting the input distribution) might not suffice even for efficiently learning depth-2 networks. The aforementioned hardness results hold already for *improper learning*, namely, where the learning algorithm is allowed to return a hypothesis that does not belong to the considered hypothesis class. Thus, a combination of assumptions on the input distribution and the network's weights seems to be necessary for obtaining efficient algorithms.

37th Conference on Neural Information Processing Systems (NeurIPS 2023).

Several polynomial-time algorithms for learning depth-2 neural networks have been obtained, under assumptions on the input distribution and the network's weights [6, 28, 43, 20, 7, 21]. In these works, it is assumed that the weight matrices are non-degenerate. That is, they either assume that the condition number of the weight matrix is bounded, or some similar non-degeneracy assumption. Specifically, Awasthi et al. [6] gave an efficient algorithm for learning depth-2 (one-hidden-layer) ReLU networks, that may include bias terms in the hidden neurons, under the assumption that the input distribution is Gaussian, and the weight matrix is non-degenerate. The non-degeneracy assumption holds w.h.p. if we add a small random noise to any weight matrix, and hence their result implies efficient learning of depth-2 ReLU networks under the Gaussian distribution in the *smoothed-analysis* framework.

The positive results on depth-2 networks suggest the following question:

> *Is there an efficient algorithm for learning ReLU networks of depth larger than* 2 *under the Gaussian distribution, where the weight matrices are non-degenerate, or in the smoothed-analysis framework where the network's parameters are smoothed?*

In this work, we give a negative answer to this question, already for depth-3 networks[1]. We show that learning depth-3 ReLU networks under the Gaussian distribution is hard even in the smoothed-analysis framework, where a random noise is added to the network's parameters. As a corollary, we show that learning depth-3 ReLU networks under the Gaussian distribution is hard even if the weight matrices are non-degenerate. Our hardness results are under a well-studied cryptographic assumption on the existence of *local pseudorandom generators (PRG)* with polynomial stretch.

Motivated by the existing positive results on smoothed-analysis in depth-2 networks, we also study whether learning depth-2 networks with smoothed parameters can be done under weak assumptions on the input distribution. Specifically, we consider the following question:

> *Is there an efficient algorithm for learning depth-2 ReLU networks in the smoothed-analysis framework, where both the network's parameters and the input distribution are smoothed?*

We give a negative answer to this question, by showing hardness of learning depth-2 ReLU networks where a random noise is added to the network's parameters, and the input distribution on $\mathbb{R}^d$ is obtained by smoothing an i.i.d. Bernoulli distribution on $\{0,1\}^d$. This hardness result is also under the assumption on the existence of local PRGs.

**Related work**

**Hardness of learning neural networks.** Hardness of improperly learning depth-2 neural networks follows from hardness of improperly learning DNFs or intersections of halfspaces, since these classes can be expressed by depth-2 networks. Klivans and Sherstov [31] showed, assuming the hardness of the shortest vector problem, that learning intersections of halfspaces is hard. Hardness of learning DNF formulas is implied by Applebaum et al. [4] under a combination of two assumptions: the first is related to the planted dense subgraph problem in hypergraphs, and the second is related to local PRGs. Daniely and Shalev-Shwartz [14] showed hardness of learning DNFs under a common assumption, namely, that refuting a random $K$-SAT formula is hard. All of the above results are distribution-free, namely, they do not imply hardness of learning neural networks under some specific distribution.

Applebaum and Raykov [3] showed, under an assumption on a specific candidate for Goldreich's PRG (i.e., based on a predicate called XOR-MAJ), that learning depth-3 Boolean circuits under the uniform distribution on the hypercube is hard. Daniely and Vardi [16] proved distribution-specific hardness of learning Boolean circuits of depth-2 (namely, DNFs) and depth-3, under the assumpion on the existence of local PRGs that we also use in this work. For DNF formulas, they showed hardness of learning under a distribution where each component is drawn i.i.d. from a Bernoulli distribution (which is non-uniform). For depth-3 Boolean circuits, they showed hardness of learning under the uniform distribution on the hypercube. Since the Boolean circuits can be expressed by ReLU networks of the same depth, these results readily translate to distribution-specific hardness of learning neural networks. Chen et al. [11] showed hardness of learning depth-2 neural networks under the uniform distribution on the hypercube, based on an assumption on the hardness of the

---

[1]We note that in our results the neural networks have the ReLU activation also in the output neuron.

*Learning with Rounding (LWR)* problem. Note that the input distributions in the above results are supported on the hypercube, and they do not immediately imply hardness of learning neural networks under continuous distributions.

When considering the computational complexity of learning neural networks, perhaps the most natural choice of an input distribution is the standard Gaussian distribution. Daniely and Vardi [16] established hardness of learning depth-3 networks under this distribution, based on the assumpion on the existence of local PRGs. Chen et al. [11] also showed hardness of learning depth-3 networks under the Gaussian distribution, but their result holds already for networks that do not have an activation function in the output neuron, and it is based on the LWR assumption. They also showed hardness of learning constant depth ReLU networks from label queries (i.e., where the learner has the ability to query the value of the target network at any desired input) under the Gaussian distribution, based either on the decisional Diffie-Hellman or the Learning with Errors assumptions.

The above results suggest that assumptions on the input distribution might not suffice for achieving an efficient algorithm for learning depth-3 neural networks. A natural question is whether assumptions on the network weights may suffice. Daniely and Vardi [15] showed (under the assumption that refuting a random $K$-SAT formula is hard) that distribution-free learning of depth-2 neural networks is hard already if the weights are drawn from some "natural" distribution or satisfy some "natural" properties. Thus, if we do not impose any assumptions on the input distribution, then even very strong assumptions on the network's weights do not suffice for efficient learning.

Several works in recent years have shown hardness of distribution-specific learning shallow neural networks using gradient-methods or statistical query (SQ) algorithms [38, 39, 41, 24, 18, 11]. It is worth noting that while the SQ framework captures some variants of the gradient-descent algorithm, it does not capture, for example, stochastic gradient-descent (SGD), which examines training points individually (see a discussion in [24]).

We emphasize that none of the above distribution-specific hardness results for neural networks (either for improper learning or SQ learning) holds in the smoothed analysis framework or for non-degenerate weights.

**Learning neural networks in polynomial time.**  Awasthi et al. [6] gave a polynomial-time algorithm for learning depth-2 (one-hidden-layer) ReLU networks, under the assumption that the input distribution is Gaussian, and the weight matrix of the target network is non-degenerate. Their algorithm is based on tensor decomposition, and it can handle bias terms in the hidden layer. Their result also implies that depth-2 ReLU networks with Gaussian inputs can be learned efficiently under the smoothed-analysis framework. Our work implies that such a result might not be possible in depth-3 networks (with activation in the output neuron). Prior to [6], several works gave polynomial time algorithms for learning depth-2 neural networks where the input distribution is Gaussian and the weight matrix is non-degenerate [28, 43, 20, 21, 7], but these works either do not handle the presence of bias terms or do not handle the ReLU activation. Some of the aforementioned works consider networks with multiple outputs, and allow certain non-Gaussian input distributions. Provable guarantees for learning neural networks in super-polynomial time were given in [18, 12, 17, 22, 41, 42, 23].

## 2   Preliminaries

**Notations.**  We use bold-face letters to denote vectors, e.g., $\mathbf{x} = (x_1, \ldots, x_d)$. For a vector $\mathbf{x}$ and a sequence $S = (i_1, \ldots, i_k)$ of $k$ indices, we let $\mathbf{x}_S = (x_{i_1}, \ldots, x_{i_k})$, i.e., the restriction of $\mathbf{x}$ to the indices $S$. We denote by $\mathbb{1}[\cdot]$ the indicator function, for example $\mathbb{1}[t \geq 5]$ equals 1 if $t \geq 5$ and 0 otherwise. For an integer $d \geq 1$ we denote $[d] = \{1, \ldots, d\}$. We denote by $\mathcal{N}(0, \sigma^2)$ the normal distribution with mean 0 and variance $\sigma^2$, and by $\mathcal{N}(\mathbf{0}, \Sigma)$ the multivariate normal distribution with mean $\mathbf{0}$ and covariance matrix $\Sigma$. The identity matrix of size $d$ is denoted by $I_d$. For $\mathbf{x} \in \mathbb{R}^d$ we denote by $\|\mathbf{x}\|$ the Euclidean norm. We use $\mathrm{poly}(a_1, \ldots, a_n)$ to denote a polynomial in $a_1, \ldots, a_n$.

**Local pseudorandom generators.**  An $(n, m, k)$-hypergraph is a hypergraph over $n$ vertices $[n]$ with $m$ hyperedges $S_1, \ldots, S_m$, each of cardinality $k$. Each hyperedge $S = (i_1, \ldots, i_k)$ is ordered, and all the $k$ members of a hyperedge are distinct. We let $\mathcal{G}_{n,m,k}$ be the distribution over such hypergraphs in which a hypergraph is chosen by picking each hyperedge uniformly and independently at random among all the possible $n \cdot (n-1) \cdot \ldots \cdot (n-k+1)$ ordered hyperedges. Let $P : \{0,1\}^k \to$

$\{0, 1\}$ be a predicate, and let $G$ be a $(n, m, k)$-hypergraph. We call *Goldreich's pseudorandom generator (PRG)* [25] the function $f_{P,G} : \{0, 1\}^n \to \{0, 1\}^m$ such that for $\mathbf{x} \in \{0, 1\}^n$, we have $f_{P,G}(\mathbf{x}) = (P(\mathbf{x}_{S_1}), \dots, P(\mathbf{x}_{S_m}))$. The integer $k$ is called the *locality* of the PRG. If $k$ is a constant then the PRG and the predicate $P$ are called *local*. We say that the PRG has *polynomial stretch* if $m = n^s$ for some constant $s > 1$. We let $\mathcal{F}_{P,n,m}$ denote the collection of functions $f_{P,G}$ where $G$ is an $(n, m, k)$-hypergraph. We sample a function from $\mathcal{F}_{P,n,m}$ by choosing a random hypergraph $G$ from $\mathcal{G}_{n,m,k}$.

We denote by $G \xleftarrow{R} \mathcal{G}_{n,m,k}$ the operation of sampling a hypergraph $G$ from $\mathcal{G}_{n,m,k}$, and by $\mathbf{x} \xleftarrow{R} \{0, 1\}^n$ the operation of sampling $\mathbf{x}$ from the uniform distribution on $\{0, 1\}^n$. We say that $\mathcal{F}_{P,n,m}$ is $\varepsilon$-pseudorandom generator ($\varepsilon$-PRG) if for every polynomial-time probabilistic algorithm $\mathcal{A}$ the *distinguishing advantage*

$$\left| \Pr_{G \xleftarrow{R} \mathcal{G}_{n,m,k}, \mathbf{x} \xleftarrow{R} \{0,1\}^n} [\mathcal{A}(G, f_{P,G}(\mathbf{x})) = 1] - \Pr_{G \xleftarrow{R} \mathcal{G}_{n,m,k}, \mathbf{y} \xleftarrow{R} \{0,1\}^m} [\mathcal{A}(G, \mathbf{y}) = 1] \right|$$

is at most $\varepsilon$. Thus, the distinguisher $\mathcal{A}$ is given a random hypergraph $G$ and a string $\mathbf{y} \in \{0, 1\}^m$, and its goal is to distinguish between the case where $\mathbf{y}$ is chosen at random, and the case where $\mathbf{y}$ is a random image of $f_{P,G}$.

Our assumption is that local PRGs with polynomial stretch and constant distinguishing advantage exist:

**Assumption 2.1.** *For every constant $s > 1$, there exists a constant $k$ and a predicate $P : \{0, 1\}^k \to \{0, 1\}$, such that $\mathcal{F}_{P,n,n^s}$ is $\frac{1}{3}$-PRG.*

We remark that the same assumption was used by Daniely and Vardi [16] to show hardness-of-learning results. Local PRGs have been extensively studied in the last two decades. In particular, local PRGs with polynomial stretch have shown to have remarkable applications, such as secure-computation with constant computational overhead [27, 5], and general-purpose obfuscation based on constant degree multilinear maps (cf. [33, 34]). Significant evidence for Assumption 2.1 was shown in Applebaum [1]. Moreover, a concrete candidate for a local PRG, based on the XOR-MAJ predicate, was shown to be secure against all known attacks [2, 13, 35, 3]. See [16] for further discussion on the assumption, and on prior work regarding the relation between Goldreich's PRG and hardness of learning.

**Neural networks.** We consider feedforward ReLU networks. Starting from an input $\mathbf{x} \in \mathbb{R}^d$, each layer in the network is of the form $\mathbf{z} \mapsto \sigma(W_i \mathbf{z} + \mathbf{b}_i)$, where $\sigma(a) = [a]_+ = \max\{0, a\}$ is the ReLU activation which applies to vectors entry-wise, $W_i$ is the weight matrix, and $\mathbf{b}_i$ are the bias terms. The *weights vector* of the $j$-th neuron in the $i$-th layer is the $j$-th row of $W_i$, and its *outgoing-weights vector* is the $j$-th column of $W_{i+1}$. We define the *depth* of the network as the number of layers. Unless stated otherwise, the output neuron also has a ReLU activation function. Note that a depth-$k$ network with activation in the output neuron has $k$ non-linear layers. A neuron which is not an input or output neuron is called a *hidden neuron*. We sometimes consider neural networks with multiple outputs. The parameters of the neural network is the set of its weight matrices and bias vectors. We often view the parameters as a vector $\boldsymbol{\theta} \in \mathbb{R}^p$ obtained by concatenating these matrices and vectors. For $B \geq 0$, we say that the parameters are $B$-bounded if the absolute values of all weights and biases are at most $B$.

**Learning neural networks and the smoothed-analysis framework.** We first define neural networks learning under the standard PAC framework:

**Definition 2.1** (Distribution-specific PAC learning). *Learning depth-$k$ neural networks under an input distribution $\mathcal{D}$ on $\mathbb{R}^d$ is defined by the following framework:*

1. *An adversary chooses a set of $B$-bounded parameters $\boldsymbol{\theta} \in \mathbb{R}^p$ for a depth-$k$ neural network $N_{\boldsymbol{\theta}} : \mathbb{R}^d \to \mathbb{R}$, as well as some $\epsilon > 0$.*

2. *Consider an examples oracle, such that each example $(\mathbf{x}, y) \in \mathbb{R}^d \times \mathbb{R}$ is drawn i.i.d. with $\mathbf{x} \sim \mathcal{D}$ and $y = N_{\boldsymbol{\theta}}(\mathbf{x})$. Then, given access to the examples oracle, the goal of the learning algorithm $\mathcal{L}$ is to return with probability at least $\frac{3}{4}$ a hypothesis $h : \mathbb{R}^d \to \mathbb{R}$ such that $\mathbb{E}_{\mathbf{x} \sim \mathcal{D}} \left[ (h(\mathbf{x}) - N_{\boldsymbol{\theta}}(\mathbf{x}))^2 \right] \leq \epsilon$. We say that $\mathcal{L}$ is* efficient *if it runs in time $\mathrm{poly}(d, p, B, 1/\epsilon)$.*

We consider learning in the *smoothed-analysis* framework [40], which is a popular paradigm for analyzing non-worst-case computational complexity [36]. The smoothed-analysis framework has been successfully applied to many learning problems (e.g., [6, 21, 30, 8, 9, 26, 19, 29, 10]). In the smoothed-analysis setting, the target network is not purely controlled by an adversary. Instead, the adversary can first generate an arbitrary network, and the parameters for this network (i.e., the weight matrices and bias terms) will be randomly perturbed to yield a perturbed network. The algorithm only needs to work with high probability on the perturbed network. This limits the power of the adversary and prevents it from creating highly degenerate cases. Formally, we consider the following framework (we note that a similar model was considered in [6, 21]):

**Definition 2.2** (Learning with smoothed parameters). *Learning depth-$k$ neural networks with smoothed parameters under an input distribution $\mathcal{D}$ is defined as follows:*

1. *An adversary chooses a set of $B$-bounded parameters $\boldsymbol{\theta} \in \mathbb{R}^p$ for a depth-$k$ neural network $N_{\boldsymbol{\theta}} : \mathbb{R}^d \to \mathbb{R}$, as well as some $\tau, \epsilon > 0$.*

2. *A perturbed set of parameters $\hat{\boldsymbol{\theta}}$ is obtained by a random perturbation to $\boldsymbol{\theta}$, namely, $\hat{\boldsymbol{\theta}} = \boldsymbol{\theta} + \boldsymbol{\xi}$ for $\boldsymbol{\xi} \sim \mathcal{N}(\mathbf{0}, \tau^2 I_p)$.*

3. *Consider an examples oracle, such that each example $(\mathbf{x}, y) \in \mathbb{R}^d \times \mathbb{R}$ is drawn i.i.d. with $\mathbf{x} \sim \mathcal{D}$ and $y = N_{\hat{\boldsymbol{\theta}}}(\mathbf{x})$. Then, given access to the examples oracle, the goal of the learning algorithm $\mathcal{L}$ is to return with probability at least $\frac{3}{4}$ (over the random perturbation $\boldsymbol{\xi}$ and the internal randomness of $\mathcal{L}$) a hypothesis $h : \mathbb{R}^d \to \mathbb{R}$ such that $\mathbb{E}_{\mathbf{x} \sim \mathcal{D}} \left[ \left( h(\mathbf{x}) - N_{\hat{\boldsymbol{\theta}}}(\mathbf{x}) \right)^2 \right] \leq \epsilon$. We say that $\mathcal{L}$ is* efficient *if it runs in time $\mathrm{poly}(d, p, B, 1/\epsilon, 1/\tau)$.*

Finally, we also consider a setting where both the parameters and the input distribution are smoothed:

**Definition 2.3** (Learning with smoothed parameters and inputs). *Learning depth-$k$ neural networks with smoothed parameters and inputs under an input distribution $\mathcal{D}$ is defined as follows:*

1. *An adversary chooses a set of $B$-bounded parameters $\boldsymbol{\theta} \in \mathbb{R}^p$ for a depth-$k$ neural network $N_{\boldsymbol{\theta}} : \mathbb{R}^d \to \mathbb{R}$, as well as some $\tau, \omega, \epsilon > 0$.*

2. *A perturbed set of parameters $\hat{\boldsymbol{\theta}}$ is obtained by a random perturbation to $\boldsymbol{\theta}$, namely, $\hat{\boldsymbol{\theta}} = \boldsymbol{\theta} + \boldsymbol{\xi}$ for $\boldsymbol{\xi} \sim \mathcal{N}(\mathbf{0}, \tau^2 I_p)$. Moreover, a smoothed input distribution $\hat{\mathcal{D}}$ is obtained from $\mathcal{D}$, such that $\hat{\mathbf{x}} \sim \hat{\mathcal{D}}$ is chosen by drawing $\mathbf{x} \sim \mathcal{D}$ and adding a random perturbation from $\mathcal{N}(\mathbf{0}, \omega^2 I_d)$.*

3. *Consider an examples oracle, such that each example $(\mathbf{x}, y) \in \mathbb{R}^d \times \mathbb{R}$ is drawn i.i.d. with $\mathbf{x} \sim \hat{\mathcal{D}}$ and $y = N_{\hat{\boldsymbol{\theta}}}(\mathbf{x})$. Then, given access to the examples oracle, the goal of the learning algorithm $\mathcal{L}$ is to return with probability at least $\frac{3}{4}$ (over the random perturbation $\boldsymbol{\xi}$ and the internal randomness of $\mathcal{L}$) a hypothesis $h : \mathbb{R}^d \to \mathbb{R}$ such that $\mathbb{E}_{\mathbf{x} \sim \hat{\mathcal{D}}} \left[ \left( h(\mathbf{x}) - N_{\hat{\boldsymbol{\theta}}}(\mathbf{x}) \right)^2 \right] \leq \epsilon$. We say that $\mathcal{L}$ is* efficient *if it runs in time $\mathrm{poly}(d, p, B, 1/\epsilon, 1/\tau, 1/\omega)$.*

We emphasize that all of the above definitions consider learning in the distribution-specific setting. Thus, the learning algorithm may depend on the specific input distribution $\mathcal{D}$.

## 3 Results

As we discussed in the introduction, there exist efficient algorithms for learning depth-2 (one-hidden-layer) ReLU networks with smoothed parameters under the Gaussian distribution. We now show that such a result may not be achieved for depth-3 networks:

**Theorem 3.1.** *Under Assumption 2.1, there is no efficient algorithm that learns depth-3 networks with smoothed parameters (in the sense of Definition 2.2) under the standard Gaussian distribution.*

We prove the theorem in Section 4. Next, we conclude that learning depth-3 neural networks under the Gaussian distribution on $\mathbb{R}^d$ is hard in the standard PAC framework even if all weight matrices are non-degenerate, namely, when the minimal singular values of the weight matrices are lower bounded by $1/\mathrm{poly}(d)$. As we discussed in the introduction, in one-hidden-layer networks with similar assumptions there exist efficient learning algorithms.

**Corollary 3.1.** *Under Assumption 2.1, there is no efficient algorithm that learns depth-3 networks (in the sense of Definition 2.1) under the standard Gaussian distribution on $\mathbb{R}^d$, even if the smallest singular value of each weight matrix is at least $1/\operatorname{poly}(d)$.*

The proof of the above corollary follows from Theorem 3.1, using the fact that by adding a small random perturbation to the weight matrices, we get w.h.p. non-degenerate matrices. Hence, an efficient algorithm that learns non-degenerate networks suffices for obtaining an efficient algorithm that learns under the smoothed analysis framework. See Appendix B for the formal proof.

The above hardness results consider depth-3 networks that include activation in the output neuron. Thus, the networks have three non-linear layers. We remark that these results readily imply hardness also for depth-4 networks without activation in the output, and hardness for depth-$k$ networks for any $k > 3$. We also note that the hardness results hold already for networks where all hidden layers are of the same width, e.g., where all layers are of width $d$.

Theorem 3.1 gives a strong limitation on learning depth-3 networks in the smoothed-analysis framework. Motivated by existing positive results on smoothed-analysis in depth-2 networks, we now study whether learning depth-2 networks with smoothed parameters can be done under weak assumptions on the input distribution. Specifically, we consider the case where both the parameters and the input distribution are smoothed. We show that efficiently learning depth-2 networks may not be possible with smoothed parameters where the input distribution on $\mathbb{R}^d$ is obtained by smoothing an i.i.d. Bernoulli distribution on $\{0,1\}^d$.

**Theorem 3.2.** *Under Assumption 2.1, there is no efficient algorithm that learns depth-2 networks with smoothed parameters and inputs (in the sense of Definition 2.3), under the distribution $\mathcal{D}$ on $\{0,1\}^d$ where each coordinate is drawn i.i.d. from a Bernoulli distribution which takes the value $0$ with probability $\frac{1}{\sqrt{d}}$.*

We prove the theorem in Appendix C. The proof follows from similar ideas to the proof of Theorem 3.1, which we discuss in the next section.

## 4 Proof of Theorem 3.1

The proof builds on a technique from Daniely and Vardi [16]. It follows by showing that an efficient algorithm for learning depth-3 neural networks with smoothed parameters under the Gaussian distribution can be used for breaking a local PRG. Intuitively, the main challenge in our proof in comparison to [16] is that our reduction must handle the random noise which is added to the parameters. Specifically, [16] define a certain examples oracle, and show that the examples returned by the oracle are realizable by some neural network which depends on the unknown $\mathbf{x} \in \{0,1\}^n$ used by the PRG. Since the network depends on this unknown $\mathbf{x}$, some of the parameters of the network are unknown, and hence it is non-trivial how to define an examples oracle which is realizable by a perturbed network. Moreover, we need to handle this random perturbation without increasing the network's depth. For these reasons, our reduction is significantly different from [16].

We now provide the formal proof. The proof relies on several lemmas which we prove in Appendix A. For a sufficiently large $n$, let $\mathcal{D}$ be the standard Gaussian distribution on $\mathbb{R}^{n^2}$. Assume that there is a $\operatorname{poly}(n)$-time algorithm $\mathcal{L}$ that learns depth-3 neural networks with at most $n^2$ hidden neurons and parameter magnitudes bounded by $n^3$, with smoothed parameters, under the distribution $\mathcal{D}$, with $\epsilon = \frac{1}{n}$, and $\tau = 1/\operatorname{poly}(n)$ that we will specify later. Let $m(n) \leq \operatorname{poly}(n)$ be the sample complexity of $\mathcal{L}$, namely, $\mathcal{L}$ uses a sample of size at most $m(n)$ and returns w.p. at least $\frac{3}{4}$ a hypothesis $h$ with $\mathbb{E}_{\mathbf{z} \sim \mathcal{D}} \left[ \left( h(\mathbf{z}) - N_{\hat{\boldsymbol{\theta}}}(\mathbf{z}) \right)^2 \right] \leq \epsilon = \frac{1}{n}$, where $N_{\hat{\boldsymbol{\theta}}}$ is the perturbed network. Let $s > 1$ be a constant such that $n^s \geq m(n) + n^3$ for every sufficiently large $n$. By Assumption 2.1, there exists a constant $k$ and a predicate $P : \{0,1\}^k \to \{0,1\}$, such that $\mathcal{F}_{P,n,n^s}$ is $\frac{1}{3}$-PRG. We will show an efficient algorithm $\mathcal{A}$ with distinguishing advantage greater than $\frac{1}{3}$ and thus reach a contradiction.

Throughout this proof, we will use the following notations. For a hyperedge $S = (i_1, \ldots, i_k)$ we denote by $\mathbf{z}^S \in \{0,1\}^{kn}$ the following encoding of $S$: the vector $\mathbf{z}^S$ is a concatenation of $k$ vectors in $\{0,1\}^n$, such that the $j$-th vector has $0$ in the $i_j$-th coordinate and $1$ elsewhere. Thus, $\mathbf{z}^S$ consists of $k$ size-$n$ slices, each encoding a member of $S$. For $\mathbf{z} \in \{0,1\}^{kn}$, $i \in [k]$ and $j \in [n]$, we denote $z_{i,j} = z_{(i-1)n+j}$. That is, $z_{i,j}$ is the $j$-th component in the $i$-th slice in $\mathbf{z}$. For $\mathbf{x} \in \{0,1\}^n$, let

$P_{\mathbf{x}} : \{0,1\}^{kn} \to \{0,1\}$ be such that for every hyperedge $S$ we have $P_{\mathbf{x}}(\mathbf{z}^S) = P(\mathbf{x}_S)$. Let $c$ be such that $\Pr_{t \sim \mathcal{N}(0,1)}[t \leq c] = \frac{1}{n}$. Let $\mu$ be the density of $\mathcal{N}(0,1)$, let $\mu_-(t) = n \cdot \mathbb{1}[t \leq c] \cdot \mu(t)$, and let $\mu_+ = \frac{n}{n-1} \cdot \mathbb{1}[t \geq c] \cdot \mu(t)$. Note that $\mu_-, \mu_+$ are the densities of the restriction of $\mu$ to the intervals $t \leq c$ and $t \geq c$ respectively. Let $\Psi : \mathbb{R}^{kn} \to \{0,1\}^{kn}$ be a mapping such that for every $\mathbf{z}' \in \mathbb{R}^{kn}$ and $i \in [kn]$ we have $\Psi(\mathbf{z}')_i = \mathbb{1}[z_i' \geq c]$. For $\tilde{\mathbf{z}} \in \mathbb{R}^{n^2}$ we denote $\tilde{\mathbf{z}}_{[kn]} = (\tilde{z}_1, \ldots, \tilde{z}_{kn})$, namely, the first $kn$ components of $\tilde{\mathbf{z}}$ (assuming $n^2 \geq kn$).

## 4.1 Defining the target network for $\mathcal{L}$

Since our goal is to use the algorithm $\mathcal{L}$ for breaking PRGs, in this subsection we define a neural network $\tilde{N} : \mathbb{R}^{n^2} \to \mathbb{R}$ that we will later use as a target network for $\mathcal{L}$. The network $\tilde{N}$ contains the subnetworks $N_1, N_2, N_3$ which we define below.

Let $N_1$ be a depth-2 neural network with input dimension $kn$, at most $n \log(n)$ hidden neurons, at most $\log(n)$ output neurons (with activations in the output neurons), and parameter magnitudes bounded by $n^3$ (all bounds are for a sufficiently large $n$), which satisfies the following. We denote the set of output neurons of $N_1$ by $\mathcal{E}_1$. Let $\mathbf{z}' \in \mathbb{R}^{kn}$ be an input to $N_1$ such that $\Psi(\mathbf{z}') = \mathbf{z}^S$ for some hyperedge $S$, and assume that for every $i \in [kn]$ we have $z_i' \notin \left(c, c + \frac{1}{n^2}\right)$. Fix some $\mathbf{x} \in \{0,1\}^n$. Then, for $S$ with $P_{\mathbf{x}}(\mathbf{z}^S) = 0$ the inputs to all output neurons $\mathcal{E}_1$ are at most $-1$, and for $S$ with $P_{\mathbf{x}}(\mathbf{z}^S) = 1$ there exists a neuron in $\mathcal{E}_1$ with input at least $2$. Recall that our definition of a neuron's input includes the addition of the bias term. The construction of the network $N_1$ is given in Lemma A.3. Intuitively, the network $N_1$ consists of a layer that transforms w.h.p. the input $\mathbf{z}'$ to $\Psi(\mathbf{z}') = \mathbf{z}^S$, followed by a layer that satisfies the following: Building on a lemma from [16] which shows that $P_{\mathbf{x}}(\mathbf{z}^S)$ can be computed by a DNF formula, we define a layer where each output neuron corresponds to a term in the DNF formula, such that if the term evaluates to $0$ then the input to the neuron is at most $-1$, and otherwise it is at least $2$. Note that the network $N_1$ depends on $\mathbf{x}$. However, only the second layer depends on $\mathbf{x}$, and thus given an input we may compute the first layer even without knowing $\mathbf{x}$. Let $N_1' : \mathbb{R}^{kn} \to \mathbb{R}$ be a depth-3 network with no activation in the output neuron, obtained from $N_1$ by summing the outputs from all neurons $\mathcal{E}_1$.

Let $N_2$ be a depth-2 neural network with input dimension $kn$, at most $n \log(n)$ hidden neurons, at most $2n$ output neurons, and parameter magnitudes bounded by $n^3$ (for a sufficiently large $n$), which satisfies the following. We denote the set of output neurons of $N_2$ by $\mathcal{E}_2$. Let $\mathbf{z}' \in \mathbb{R}^{kn}$ be an input to $N_2$ such that for every $i \in [kn]$ we have $z_i' \notin \left(c, c + \frac{1}{n^2}\right)$. If $\Psi(\mathbf{z}')$ is an encoding of a hyperedge then the inputs to all output neurons $\mathcal{E}_2$ are at most $-1$, and otherwise there exists a neuron in $\mathcal{E}_2$ with input at least $2$. The construction of the network $N_2$ is given in Lemma A.5. Intuitively, each neuron in $\mathcal{E}_2$ is responsible for checking whether $\Psi(\mathbf{z}')$ violates some requirement that must hold in an encoding of a hyperedge. Let $N_2' : \mathbb{R}^{kn} \to \mathbb{R}$ be a depth-3 network with no activation in the output neuron, obtained from $N_2$ by summing the outputs from all neurons $\mathcal{E}_2$.

Let $N_3$ be a depth-2 neural network with input dimension $kn$, at most $n \log(n)$ hidden neurons, $kn \leq n \log(n)$ output neurons, and parameter magnitudes bounded by $n^3$ (for a sufficiently large $n$), which satisfies the following. We denote the set of output neurons of $N_3$ by $\mathcal{E}_3$. Let $\mathbf{z}' \in \mathbb{R}^{kn}$ be an input to $N_3$. If there exists $i \in [kn]$ such that $z_i' \in \left(c, c + \frac{1}{n^2}\right)$ then there exists a neuron in $\mathcal{E}_3$ with input at least $2$. Moreover, if for all $i \in [kn]$ we have $z_i' \notin \left(c - \frac{1}{n^2}, c + \frac{2}{n^2}\right)$ then the inputs to all neurons in $\mathcal{E}_3$ are at most $-1$. The construction of the network $N_3$ is straightforward and given in Lemma A.6. Let $N_3' : \mathbb{R}^{kn} \to \mathbb{R}$ be a depth-3 neural network with no activation function in the output neuron, obtained from $N_3$ by summing the outputs from all neurons $\mathcal{E}_3$.

Let $N' : \mathbb{R}^{kn} \to \mathbb{R}$ be a depth-3 network obtained from $N_1', N_2', N_3'$ as follows. For $\mathbf{z}' \in \mathbb{R}^{kn}$ we have $N'(\mathbf{z}') = [1 - N_1'(\mathbf{z}') - N_2'(\mathbf{z}') - N_3'(\mathbf{z}')]_+$. The network $N'$ has at most $n^2$ neurons, and parameter magnitudes bounded by $n^3$ (all bounds are for a sufficiently large $n$). Finally, let $\tilde{N} : \mathbb{R}^{n^2} \to \mathbb{R}$ be a depth-3 neural network such that $\tilde{N}(\tilde{\mathbf{z}}) = N'\left(\tilde{\mathbf{z}}_{[kn]}\right)$.

## 4.2 Defining the noise magnitude $\tau$ and analyzing the perturbed network

In order to use the algorithm $\mathcal{L}$ w.r.t. some neural network with parameters $\boldsymbol{\theta}$, we need to implement an examples oracle, such that the examples are labeled according to a neural network with parameters $\boldsymbol{\theta} + \boldsymbol{\xi}$, where $\boldsymbol{\xi}$ is a random perturbation. Specifically, we use $\mathcal{L}$ with an examples oracle where the

labels correspond to a network $\hat{N}: \mathbb{R}^{n^2} \to \mathbb{R}$, obtained from $\tilde{N}$ (w.r.t. an appropriate $\mathbf{x} \in \{0,1\}^n$ in the construction of $N_1$) by adding a small perturbation to the parameters. The perturbation is such that we add i.i.d. noise to each parameter in $\tilde{N}$, where the noise is distributed according to $\mathcal{N}(0, \tau^2)$, and $\tau = 1/\operatorname{poly}(n)$ is small enough such that the following holds. Let $f_{\boldsymbol{\theta}}: \mathbb{R}^{n^2} \to \mathbb{R}$ be any depth-3 neural network parameterized by $\boldsymbol{\theta} \in \mathbb{R}^r$ for some $r > 0$ with at most $n^2$ neurons, and parameter magnitudes bounded by $n^3$ (note that $r$ is polynomial in $n$). Then w.p. at least $1 - \frac{1}{n}$ over $\boldsymbol{\xi} \sim \mathcal{N}(\mathbf{0}, \tau^2 I_r)$, we have $|\xi_i| \leq \frac{1}{10}$ for all $i \in [r]$, and the network $f_{\boldsymbol{\theta}+\boldsymbol{\xi}}$ is such that for every input $\tilde{\mathbf{z}} \in \mathbb{R}^{n^2}$ with $\|\tilde{\mathbf{z}}\| \leq 2n$ and every neuron we have: Let $a, b$ be the inputs to the neuron in the computations $f_{\boldsymbol{\theta}}(\tilde{\mathbf{z}})$ and $f_{\boldsymbol{\theta}+\boldsymbol{\xi}}(\tilde{\mathbf{z}})$ (respectively), then $|a - b| \leq \frac{1}{2}$. Thus, $\tau$ is sufficiently small, such that w.h.p. adding i.i.d. noise $\mathcal{N}(0, \tau^2)$ to each parameter does not change the inputs to the neurons by more than $\frac{1}{2}$. Note that such an inverse-polynomial $\tau$ exists, since when the network size, parameter magnitudes, and input size are bounded by some $\operatorname{poly}(n)$, then the input to each neuron in $f_{\boldsymbol{\theta}}(\tilde{\mathbf{z}})$ is $\operatorname{poly}(n)$-Lipschitz as a function of $\boldsymbol{\theta}$, and thus it suffices to choose $\tau$ that implies w.p. at least $1 - \frac{1}{n}$ that $\|\boldsymbol{\xi}\| \leq \frac{1}{q(n)}$ for a sufficiently large polynomial $q(n)$ (see Lemma A.7 for details).

Let $\tilde{\boldsymbol{\theta}} \in \mathbb{R}^p$ be the parameters of the network $\tilde{N}$. Recall that the parameters vector $\tilde{\boldsymbol{\theta}}$ is the concatenation of all weight matrices and bias terms. Let $\hat{\boldsymbol{\theta}} \in \mathbb{R}^p$ be the parameters of $\hat{N}$, namely, $\hat{\boldsymbol{\theta}} = \tilde{\boldsymbol{\theta}} + \boldsymbol{\xi}$ where $\boldsymbol{\xi} \sim \mathcal{N}(\mathbf{0}, \tau^2 I_p)$. By our choice of $\tau$ and the construction of the networks $N_1, N_2, N_3$, w.p. at least $1 - \frac{1}{n}$ over $\boldsymbol{\xi}$, for every $\tilde{\mathbf{z}}$ with $\|\tilde{\mathbf{z}}\| \leq 2n$, the inputs to the neurons $\mathcal{E}_1, \mathcal{E}_2, \mathcal{E}_3$ in the computation $\hat{N}(\tilde{\mathbf{z}})$ satisfy the following properties, where we denote $\mathbf{z}' = \tilde{\mathbf{z}}_{[kn]}$:

(P1) If $\Psi(\mathbf{z}') = \mathbf{z}^S$ for some hyperedge $S$, and for every $i \in [kn]$ we have $z_i' \notin \left(c, c + \frac{1}{n^2}\right)$, then the inputs to $\mathcal{E}_1$ satisfy: If $P_{\mathbf{x}}(\mathbf{z}^S) = 0$ then the inputs to all neurons in $\mathcal{E}_1$ are at most $-\frac{1}{2}$, and if $P_{\mathbf{x}}(\mathbf{z}^S) = 1$ then there exists a neuron in $\mathcal{E}_1$ with input at least $\frac{3}{2}$.

(P2) If for every $i \in [kn]$ we have $z_i' \notin \left(c, c + \frac{1}{n^2}\right)$, then the inputs to $\mathcal{E}_2$ satisfy: If $\Psi(\mathbf{z}')$ is an encoding of a hyperedge then the inputs to all neurons $\mathcal{E}_2$ are at most $-\frac{1}{2}$, and otherwise there exists a neuron in $\mathcal{E}_2$ with input at least $\frac{3}{2}$.

(P3) The inputs to $\mathcal{E}_3$ satisfy: If there exists $i \in [kn]$ such that $z_i' \in \left(c, c + \frac{1}{n^2}\right)$ then there exists a neuron in $\mathcal{E}_3$ with input at least $\frac{3}{2}$, and if for all $i \in [kn]$ we have $z_i' \notin \left(c - \frac{1}{n^2}, c + \frac{2}{n^2}\right)$ then the inputs to all neurons in $\mathcal{E}_3$ are at most $-\frac{1}{2}$.

### 4.3 Stating the algorithm $\mathcal{A}$

Given a sequence $(S_1, y_1), \ldots, (S_{n^s}, y_{n^s})$, where $S_1, \ldots, S_{n^s}$ are i.i.d. random hyperedges, the algorithm $\mathcal{A}$ needs to distinguish whether $\mathbf{y} = (y_1, \ldots, y_{n^s})$ is random or that we have $\mathbf{y} = (P(\mathbf{x}_{S_1}), \ldots, P(\mathbf{x}_{S_{n^s}})) = (P_{\mathbf{x}}(\mathbf{z}^{S_1}), \ldots, P_{\mathbf{x}}(\mathbf{z}^{S_{n^s}}))$ for a random $\mathbf{x} \in \{0,1\}^n$. We let $\mathcal{S} = ((\mathbf{z}^{S_1}, y_1), \ldots, (\mathbf{z}^{S_{n^s}}, y_{n^s}))$.

We use the efficient algorithm $\mathcal{L}$ in order to obtain distinguishing advantage greater than $\frac{1}{3}$ as follows. Let $\boldsymbol{\xi}$ be a random perturbation, and let $\hat{N}$ be the perturbed network as defined above, w.r.t. the unknown $\mathbf{x} \in \{0,1\}^n$. Note that given a perturbation $\boldsymbol{\xi}$, only the weights in the second layer of the subnetwork $N_1$ in $\hat{N}$ are unknown, since all other parameters do not depend on $\mathbf{x}$. The algorithm $\mathcal{A}$ runs $\mathcal{L}$ with the following examples oracle. In the $i$-th call, the oracle first draws $\mathbf{z} \in \{0,1\}^{kn}$ such that each component is drawn i.i.d. from a Bernoulli distribution which takes the value 0 w.p. $\frac{1}{n}$. If $\mathbf{z}$ is an encoding of a hyperedge then the oracle replaces $\mathbf{z}$ with $\mathbf{z}^{S_i}$. Then, the oracle chooses $\mathbf{z}' \in \mathbb{R}^{kn}$ such that for each component $j$, if $z_j = 1$ then $z_j'$ is drawn from $\mu_+$, and otherwise $z_j'$ is drawn from $\mu_-$. Let $\tilde{\mathbf{z}} \in \mathbb{R}^{n^2}$ be such that $\tilde{\mathbf{z}}_{[kn]} = \mathbf{z}'$, and the other $n^2 - kn$ components of $\tilde{\mathbf{z}}$ are drawn i.i.d. from $\mathcal{N}(0, 1)$. Note that the vector $\tilde{\mathbf{z}}$ has the distribution $\mathcal{D}$, due to the definitions of the densities $\mu_+$ and $\mu_-$, and since replacing an encoding of a random hyperedge by an encoding of another random hyperedge does not change the distribution of $\mathbf{z}$. Let $\hat{b} \in \mathbb{R}$ be the bias term of the output neuron of $\hat{N}$. The oracle returns $(\tilde{\mathbf{z}}, \tilde{y})$, where the labels $\tilde{y}$ are chosen as follows:

- If $\Psi(\mathbf{z}')$ is not an encoding of a hyperedge, then $\tilde{y} = 0$.

- If $\Psi(\mathbf{z}')$ is an encoding of a hyperedge:

- If $\mathbf{z}'$ does not have components in the interval $(c - \frac{1}{n^2}, c + \frac{2}{n^2})$, then if $y_i = 0$ we set $\tilde{y} = \hat{b}$, and if $y_i = 1$ we set $\tilde{y} = 0$.

- If $\mathbf{z}'$ has a component in the interval $(c, c + \frac{1}{n^2})$, then $\tilde{y} = 0$.

- If $\mathbf{z}'$ does not have components in the interval $(c, c + \frac{1}{n^2})$, but has a component in the interval $(c - \frac{1}{n^2}, c + \frac{2}{n^2})$, then the label $\tilde{y}$ is determined as follows: If $y_i = 1$ then $\tilde{y} = 0$. If $y_i = 0$, we have: Let $\hat{N}_3$ be the network $\hat{N}$ after omitting the neurons $\mathcal{E}_1, \mathcal{E}_2$ and their incoming and outgoing weights. Then, we set $\tilde{y} = [\hat{b} - \hat{N}_3(\tilde{\mathbf{z}})]_+$. Note that since only the second layer of $N_1$ depends on $\mathbf{x}$, then we can compute $\hat{N}_3(\tilde{\mathbf{z}})$ without knowing $\mathbf{x}$.

Let $h$ be the hypothesis returned by $\mathcal{L}$. Recall that $\mathcal{L}$ uses at most $m(n)$ examples, and hence $\mathcal{S}$ contains at least $n^3$ examples that $\mathcal{L}$ cannot view. We denote the indices of these examples by $I = \{m(n) + 1, \ldots, m(n) + n^3\}$, and the examples by $\mathcal{S}_I = \{(\mathbf{z}^{S_i}, y_i)\}_{i \in I}$. By $n^3$ additional calls to the oracle, the algorithm $\mathcal{A}$ obtains the examples $\tilde{\mathcal{S}}_I = \{(\tilde{\mathbf{z}}_i, \tilde{y}_i)\}_{i \in I}$ that correspond to $\mathcal{S}_I$. Let $h'$ be a hypothesis such that for all $\tilde{\mathbf{z}} \in \mathbb{R}^{n^2}$ we have $h'(\tilde{\mathbf{z}}) = \max\{0, \min\{\hat{b}, h(\tilde{\mathbf{z}})\}\}$, thus, for $\hat{b} \geq 0$ the hypothesis $h'$ is obtained from $h$ by clipping the output to the interval $[0, \hat{b}]$. Let $\ell_I(h') = \frac{1}{|I|} \sum_{i \in I} (h'(\tilde{\mathbf{z}}_i) - \tilde{y}_i)^2$. Now, if $\ell_I(h') \leq \frac{2}{n}$, then $\mathcal{A}$ returns 1, and otherwise it returns 0. We remark that the decision of our algorithm is based on $h'$ (rather than $h$) since we need the outputs to be bounded, in order to allow using Hoeffding's inequality in our analysis, which we discuss in the next subsection.

## 4.4 Analyzing the algorithm $\mathcal{A}$

Note that the algorithm $\mathcal{A}$ runs in $\mathrm{poly}(n)$ time. We now show that if $\mathcal{S}$ is pseudorandom then $\mathcal{A}$ returns 1 w.p. greater than $\frac{2}{3}$, and if $\mathcal{S}$ is random then $\mathcal{A}$ returns 1 w.p. less than $\frac{1}{3}$.

We start with the case where $\mathcal{S}$ is pseudorandom. In Lemma A.8, we prove that if $\mathcal{S}$ is pseudorandom then w.h.p. (over $\boldsymbol{\xi} \sim \mathcal{N}(\mathbf{0}, \tau^2 I_p)$ and the i.i.d. inputs $\tilde{\mathbf{z}}_i \sim \mathcal{D}$) the examples $(\tilde{\mathbf{z}}_1, \tilde{y}_1), \ldots, (\tilde{\mathbf{z}}_{m(n)+n^3}, \tilde{y}_{m(n)+n^3})$ returned by the oracle are realized by $\hat{N}$. Thus, $\tilde{y}_i = \hat{N}(\tilde{\mathbf{z}}_i)$ for all $i$. As we show in the lemma, this claim follows by considering the definition of the oracle for the different cases of $(\tilde{\mathbf{z}}_i)_{[kn]}$, and using Properties (P1)–(P3) to show that $\hat{N}$ behaves similarly.

Recall that the algorithm $\mathcal{L}$ is such that w.p. at least $\frac{3}{4}$ (over $\boldsymbol{\xi} \sim \mathcal{N}(\mathbf{0}, \tau^2 I_p)$, the i.i.d. inputs $\tilde{\mathbf{z}}_i \sim \mathcal{D}$, and its internal randomness), given a size-$m(n)$ dataset labeled by $\hat{N}$, it returns a hypothesis $h$ such that $\mathbb{E}_{\tilde{\mathbf{z}} \sim \mathcal{D}} \left[ (h(\tilde{\mathbf{z}}) - \hat{N}(\tilde{\mathbf{z}}))^2 \right] \leq \frac{1}{n}$. By the definition of $h'$ and the construction of $\hat{N}$, if $h$ has small error then $h'$ also has small error, namely, we have $\mathbb{E}_{\tilde{\mathbf{z}} \sim \mathcal{D}} \left[ (h'(\tilde{\mathbf{z}}) - \hat{N}(\tilde{\mathbf{z}}))^2 \right] \leq \frac{1}{n}$. In Lemma A.9 we use the above arguments and Hoeffding's inequality over $\tilde{\mathcal{S}}_I$, and prove that w.p. greater than $\frac{2}{3}$ we have $\ell_I(h') \leq \frac{2}{n}$.

Next, we consider the case where $\mathcal{S}$ is random. Let $\tilde{\mathcal{Z}} \subseteq \mathbb{R}^{n^2}$ be such that $\tilde{\mathbf{z}} \in \tilde{\mathcal{Z}}$ iff $\tilde{\mathbf{z}}_{[kn]}$ does not have components in the interval $(c - \frac{1}{n^2}, c + \frac{2}{n^2})$, and $\Psi(\tilde{\mathbf{z}}_{[kn]}) = \mathbf{z}^S$ for a hyperedge $S$. If $\mathcal{S}$ is random, then by the definition of our examples oracle, for every $i \in [m(n) + n^3]$ such that $\tilde{\mathbf{z}}_i \in \tilde{\mathcal{Z}}$, we have $\tilde{y}_i = \hat{b}$ w.p. $\frac{1}{2}$ and $\tilde{y}_i = 0$ otherwise. Also, by the definition of the oracle, $\tilde{y}_i$ is independent of $S_i$ and independent of the choice of the vector $\tilde{\mathbf{z}}_i$ that corresponds to $\mathbf{z}^{S_i}$. Hence, for such $\tilde{\mathbf{z}}_i \in \tilde{\mathcal{Z}}$ with $i \in I$, any hypothesis cannot predict the label $\tilde{y}_i$, and the expected loss for the example is at least $\left( \frac{\hat{b}}{2} \right)^2$. Moreover, in Lemma A.11 we show that $\Pr \left[ \tilde{\mathbf{z}}_i \in \tilde{\mathcal{Z}} \right] \geq \frac{1}{2 \log(n)}$ for a sufficiently large $n$. In Lemma A.12 we use these arguments to prove a lower bound on $\mathbb{E}_{\tilde{\mathcal{S}}_I} [\ell_I(h')]$, and by Hoeffding's inequality over $\tilde{\mathcal{S}}_I$ we conclude that w.p. greater than $\frac{2}{3}$ we have $\ell_I(h') > \frac{2}{n}$.

Overall, if $\mathcal{S}$ is pseudorandom then w.p. greater than $\frac{2}{3}$ the algorithm $\mathcal{A}$ returns 1, and if $\mathcal{S}$ is random then w.p. greater than $\frac{2}{3}$ it returns 0. Thus, the distinguishing advantage is greater than $\frac{1}{3}$.

# 5  Discussion

Understanding the computational complexity of learning neural networks is a central question in learning theory. Our results imply that the assumptions which allow for efficient learning in one-hidden-layer networks might not suffice in deeper networks. Also, in depth-2 networks we show that it is not sufficient to assume that both the parameters and the inputs are smoothed. We hope that our hardness results will help focus on assumptions that may allow for efficient learning.

We emphasize that our hardness results are for neural networks that include the ReLU activation also in the output neuron. In contrast, the positive results on learning depth-2 networks that we discussed in the introduction do not include activation in the output neuron. Therefore, as far as we are aware, there is still a small gap between the upper bounds and our hardness results: (1) Under the assumption that the input is Gaussian and the weights are non-degenerate, the cases of depth-2 networks with activation in the output neuron and of depth-3 networks without activation in the output are not settled; (2) In the setting where both the parameters and the input distribution are smoothed, the case of depth-2 networks without activation in the output is not settled. These gaps are an intriguing subject for further research.

## Acknowledgments and Disclosure of Funding

This work was done as part of the NSF-Simons Sponsored Collaboration on the Theoretical Foundations of Deep Learning.

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

# A    Missing lemmas for the proof of Theorem 3.1

**Lemma A.1** (Daniely and Vardi [16]). *For every predicate $P : \{0,1\}^k \to \{0,1\}$ and $\mathbf{x} \in \{0,1\}^n$, there is a DNF formula $\psi$ over $\{0,1\}^{kn}$ with at most $2^k$ terms, such that for every hyperedge $S$ we have $P_{\mathbf{x}}(\mathbf{z}^S) = \psi(\mathbf{z}^S)$. Moreover, each term in $\psi$ is a conjunction of positive literals.*

*Proof.* The following proof is from Daniely and Vardi [16], and we give it here for completeness.

We denote by $\mathcal{B} \subseteq \{0,1\}^k$ the set of satisfying assignments of $P$. Note that the size of $\mathcal{B}$ is at most $2^k$. Consider the following DNF formula over $\{0,1\}^{kn}$:

$$\psi(\mathbf{z}) = \bigvee_{\mathbf{b} \in \mathcal{B}} \bigwedge_{j \in [k]} \bigwedge_{\{l : x_l \neq b_j\}} z_{j,l} \,.$$

For a hyperedge $S = (i_1, \ldots, i_k)$, we have

$$
\begin{aligned}
\psi(\mathbf{z}^S) = 1 &\iff \exists \mathbf{b} \in \mathcal{B} \; \forall j \in [k] \; \forall x_l \neq b_j, \; z_{j,l}^S = 1 \\
&\iff \exists \mathbf{b} \in \mathcal{B} \; \forall j \in [k] \; \forall x_l \neq b_j, \; i_j \neq l \\
&\iff \exists \mathbf{b} \in \mathcal{B} \; \forall j \in [k], \; x_{i_j} = b_j \\
&\iff \exists \mathbf{b} \in \mathcal{B}, \; \mathbf{x}_S = \mathbf{b} \\
&\iff P(\mathbf{x}_S) = 1 \\
&\iff P_{\mathbf{x}}(\mathbf{z}^S) = 1 \,.
\end{aligned}
$$

$\square$

**Lemma A.2.** *Let $\mathbf{x} \in \{0,1\}^n$. There exists an affine layer with at most $2^k$ outputs, weights bounded by a constant and bias terms bounded by $n \log(n)$ (for a sufficiently large $n$), such that given an input $\mathbf{z}^S \in \{0,1\}^{kn}$ for some hyperedge $S$, it satisfies the following: For $S$ with $P_{\mathbf{x}}(\mathbf{z}^S) = 0$ all outputs are at most $-1$, and for $S$ with $P_{\mathbf{x}}(\mathbf{z}^S) = 1$ there exists an output greater or equal to $2$.*

*Proof.* By Lemma A.1, there exists a DNF formula $\varphi_{\mathbf{x}}$ over $\{0,1\}^{kn}$ with at most $2^k$ terms, such that $\varphi_{\mathbf{x}}(\mathbf{z}^S) = P_{\mathbf{x}}(\mathbf{z}^S)$. Thus, if $P_{\mathbf{x}}(\mathbf{z}^S) = 0$ then all terms in $\varphi_{\mathbf{x}}$ are not satisfied for the input $\mathbf{z}^S$, and if $P_{\mathbf{x}}(\mathbf{z}^S) = 1$ then there is at least one term in $\varphi_{\mathbf{x}}$ which is satisfied for the input $\mathbf{z}^S$. Therefore, it suffices to construct an affine layer such that for an input $\mathbf{z}^S$, the $j$-th output will be at most $-1$ if the $j$-th term of $\varphi_{\mathbf{x}}$ is not satisfied, and at least $2$ otherwise. Each term $C_j$ in $\varphi_{\mathbf{x}}$ is a conjunction of positive literals. Let $I_j \subseteq [kn]$ be the indices of these literals. The $j$-th output of the affine layer will be

$$\left( \sum_{l \in I_j} 3 z_l^S \right) - 3|I_j| + 2 \,.$$

Note that if the conjunction $C_j$ holds, then this expression is exactly $3|I_j| - 3|I_j| + 2 = 2$, and otherwise it is at most $3(|I_j| - 1) - 3|I_j| + 2 = -1$. Finally, note that all weights are bounded by $3$ and all bias terms are bounded by $n \log(n)$ (for large enough $n$). $\square$

**Lemma A.3.** *Let $\mathbf{x} \in \{0,1\}^n$. There exists a depth-2 neural network $N_1$ with input dimension $kn$, $2kn$ hidden neurons, at most $2^k$ output neurons, and parameter magnitudes bounded by $n^3$ (for a sufficiently large $n$), which satisfies the following. We denote the set of output neurons of $N_1$ by $\mathcal{E}_1$. Let $\mathbf{z}' \in \mathbb{R}^{kn}$ be such that $\Psi(\mathbf{z}') = \mathbf{z}^S$ for some hyperedge $S$, and assume that for every $i \in [kn]$ we have $z_i' \notin \left( c, c + \frac{1}{n^2} \right)$. Then, for $S$ with $P_{\mathbf{x}}(\mathbf{z}^S) = 0$ the inputs to all neurons $\mathcal{E}_1$ are at most $-1$, and for $S$ with $P_{\mathbf{x}}(\mathbf{z}^S) = 1$ there exists a neuron in $\mathcal{E}_1$ with input at least $2$. Moreover, only the second layer of $N_1$ depends on $\mathbf{x}$.*

*Proof.* First, we construct a depth-2 neural network $N_\Psi : \mathbb{R}^{kn} \to [0,1]^{kn}$ with a single layer of non-linearity, such that for every $\mathbf{z}' \in \mathbb{R}^{kn}$ with $z_i' \notin (c, c + \frac{1}{n^2})$ for every $i \in [kn]$, we have $N_\Psi(\mathbf{z}') = \Psi(\mathbf{z}')$. The network $N_\Psi$ has $2kn$ hidden neurons, and computes $N_\Psi(\mathbf{z}') = (f(z_1'), \ldots, f(z_{kn}'))$, where $f : \mathbb{R} \to [0,1]$ is such that

$$f(t) = n^2 \cdot \left( [t - c]_+ - \left[ t - \left( c + \frac{1}{n^2} \right) \right]_+ \right) \,.$$

Note that if $t \leq c$ then $f(t) = 0$, if $t \geq c + \frac{1}{n^2}$ then $f(t) = 1$, and if $c < t < c + \frac{1}{n^2}$ then $f(t) \in (0, 1)$. Also, note that all weights and bias terms can be bounded by $n^2$ (for large enough $n$). Moreover, the network $N_\Psi$ does not depend on $\mathbf{x}$.

Let $\mathbf{z}' \in \mathbb{R}^{kn}$ such that $\Psi(\mathbf{z}') = \mathbf{z}^S$ for some hyperedge $S$, and assume that for every $i \in [kn]$ we have $z_i' \notin \left(c, c + \frac{1}{n^2}\right)$. For such $\mathbf{z}'$, we have $N_\Psi(\mathbf{z}') = \Psi(\mathbf{z}') = \mathbf{z}^S$. Hence, it suffices to show that we can construct an affine layer with at most $2^k$ outputs, weights bounded by a constant and bias terms bounded by $n^3$, such that given an input $\mathbf{z}^S$ it satisfies the following: For $S$ with $P_\mathbf{x}(\mathbf{z}^S) = 0$ all outputs are at most $-1$, and for $S$ with $P_\mathbf{x}(\mathbf{z}^S) = 1$ there exists an output greater or equal to $2$. We construct such an affine layer in Lemma A.2. $\qquad\square$

**Lemma A.4.** *There exists an affine layer with $2k + n$ outputs, weights bounded by a constant and bias terms bounded by $n \log(n)$ (for a sufficiently large $n$), such that given an input $\mathbf{z} \in \{0, 1\}^{kn}$, if it is an encoding of a hyperedge then all outputs are at most $-1$, and otherwise there exists an output greater or equal to $2$.*

*Proof.* Note that $\mathbf{z} \in \{0, 1\}^{kn}$ is not an encoding of a hyperedge iff at least one of the following holds:

1. At least one of the $k$ size-$n$ slices in $\mathbf{z}$ contains $0$ more than once.

2. At least one of the $k$ size-$n$ slices in $\mathbf{z}$ does not contain $0$.

3. There are two size-$n$ slices in $\mathbf{z}$ that encode the same index.

We define the outputs of our affine layer as follows. First, we have $k$ outputs that correspond to (1). In order to check whether slice $i \in [k]$ contains $0$ more than once, the output will be $3n - 4 - \left(\sum_{j \in [n]} 3z_{i,j}\right)$. Second, we have $k$ outputs that correspond to (2): in order to check whether slice $i \in [k]$ does not contain $0$, the output will be $\left(\sum_{j \in [n]} 3z_{i,j}\right) - 3n + 2$. Finally, we have $n$ outputs that correspond to (3): in order to check whether there are two slices that encode the same index $j \in [n]$, the output will be $3k - 4 - \left(\sum_{i \in [k]} 3z_{i,j}\right)$. Note that all weights are bounded by $3$ and all bias terms are bounded by $n \log(n)$ for large enought $n$. $\qquad\square$

**Lemma A.5.** *There exists a depth-$2$ neural network $N_2$ with input dimension $kn$, at most $2kn$ hidden neurons, $2k + n$ output neurons, and parameter magnitudes bounded by $n^3$ (for a sufficiently large $n$), which satisfies the following. We denote the set of output neurons of $N_2$ by $\mathcal{E}_2$. Let $\mathbf{z}' \in \mathbb{R}^{kn}$ be such that for every $i \in [kn]$ we have $z_i' \notin \left(c, c + \frac{1}{n^2}\right)$. If $\Psi(\mathbf{z}')$ is an encoding of a hyperedge then the inputs to all neurons $\mathcal{E}_2$ are at most $-1$, and otherwise there exists a neuron in $\mathcal{E}_2$ with input at least $2$.*

*Proof.* Let $N_\Psi : \mathbb{R}^{kn} \to [0, 1]^{kn}$ be the depth-$2$ neural network from the proof of Lemma A.3, with a single layer of non-linearity with $2kn$ hidden neurons, and parameter magnitudes bounded by $n^2$, such that for every $\mathbf{z}' \in \mathbb{R}^{kn}$ with $z_i' \notin (c, c + \frac{1}{n^2})$ for every $i \in [kn]$, we have $N_\Psi(\mathbf{z}') = \Psi(\mathbf{z}')$.

Let $\mathbf{z}' \in \mathbb{R}^{kn}$ be such that for every $i \in [kn]$ we have $z_i' \notin \left(c, c + \frac{1}{n^2}\right)$. For such $\mathbf{z}'$ we have $N_\Psi(\mathbf{z}') = \Psi(\mathbf{z}')$. Hence, it suffices to show that we can construct an affine layer with $2k + n$ outputs, weights bounded by a constant and bias terms bounded by $n^3$, such that given an input $\mathbf{z} \in \{0, 1\}^{kn}$, if it is an encoding of a hyperedge then all outputs are at most $-1$, and otherwise there exists an output greater or equal to $2$. We construct such an affine layer in Lemma A.4. $\qquad\square$

**Lemma A.6.** *There exists a depth-$2$ neural network $N_3$ with input dimension $kn$, at most $n \log(n)$ hidden neurons, $kn \leq n \log(n)$ output neurons, and parameter magnitudes bounded by $n^3$ (for a sufficiently large $n$), which satisfies the following. We denote the set of output neurons of $N_3$ by $\mathcal{E}_3$. Let $\mathbf{z}' \in \mathbb{R}^{kn}$. If there exists $i \in [kn]$ such that $z_i' \in \left(c, c + \frac{1}{n^2}\right)$ then there exists a neuron in $\mathcal{E}_3$ with input at least $2$. If for all $i \in [kn]$ we have $z_i' \notin \left(c - \frac{1}{n^2}, c + \frac{2}{n^2}\right)$ then the inputs to all neurons in $\mathcal{E}_3$ are at most $-1$.*

*Proof.* It suffices to construct a univariate depth-$2$ network $f : \mathbb{R} \to \mathbb{R}$ with one non-linear layer and a constant number of hidden neurons, such that for every input $z_i' \in (c, c + \frac{1}{n^2})$ we have $f(z_i') = 2$, and for every $z_i' \notin (c - \frac{1}{n^2}, c + \frac{2}{n^2})$ we have $f(z_i') = -1$.

We construct $f$ as follows:

$$f(z_i') = (3n^2) \left( \left[ z_i' - \left( c - \frac{1}{n^2} \right) \right]_+ - [z_i' - c]_+ \right) -$$

$$(3n^2) \left( \left[ z_i' - \left( c + \frac{1}{n^2} \right) \right]_+ - \left[ z_i' - \left( c + \frac{2}{n^2} \right) \right]_+ \right) - 1 \ .$$

Note that all weights and bias terms are bounded by $n^3$ (for large enough $n$). $\qquad\square$

**Lemma A.7.** *Let $q = \mathrm{poly}(n)$ and $r = \mathrm{poly}(n)$. Then, there exists $\tau = \frac{1}{\mathrm{poly}(n)}$ such that for a sufficiently large $n$, with probability at least $1 - \exp(-n/2)$ a vector $\boldsymbol{\xi} \sim \mathcal{N}(\mathbf{0}, \tau^2 I_r)$ satisfies $\|\boldsymbol{\xi}\| \leq \frac{1}{q}$.*

*Proof.* Let $\tau = \frac{1}{q\sqrt{2rn}}$. Every component $\xi_i$ in $\boldsymbol{\xi}$ has the distribution $\mathcal{N}(0, \tau^2)$. By a standard tail bound of the Gaussian distribution, we have for every $i \in [r]$ and $t \geq 0$ that $\Pr[\xi_i \geq t] \leq 2\exp\left( -\frac{t^2}{2\tau^2} \right)$. Hence, for $t = \frac{1}{q\sqrt{r}}$, we get

$$\Pr\left[ \xi_i \geq \frac{1}{q\sqrt{r}} \right] \leq 2\exp\left( -\frac{1}{2\tau^2 q^2 r} \right) = 2\exp\left( -\frac{2rnq^2}{2q^2 r} \right) = 2\exp\left( -n \right) \ .$$

By the union bound, with probability at least $1 - r \cdot 2e^{-n}$, we have

$$\|\boldsymbol{\xi}\|^2 \leq r \cdot \frac{1}{rq^2} = \frac{1}{q^2} \ .$$

Thus, for a sufficiently large $n$, with probability at least $1 - \exp(-n/2)$ we have $\|\boldsymbol{\xi}\| \leq \frac{1}{q}$. $\qquad\square$

**Lemma A.8.** *If $\mathcal{S}$ is pseudorandom then with probability at least $\frac{39}{40}$ (over $\boldsymbol{\xi} \sim \mathcal{N}(\mathbf{0}, \tau^2 I_p)$ and the i.i.d. inputs $\tilde{\mathbf{z}}_i \sim \mathcal{D}$) the examples $(\tilde{\mathbf{z}}_1, \tilde{y}_1), \ldots, (\tilde{\mathbf{z}}_{m(n)+n^3}, \tilde{y}_{m(n)+n^3})$ returned by the oracle are realized by $\hat{N}$.*

*Proof.* By our choice of $\tau$, with probability at least $1 - \frac{1}{n}$ over $\boldsymbol{\xi} \sim \mathcal{N}(\mathbf{0}, \tau^2 I_p)$, we have $|\xi_j| \leq \frac{1}{10}$ for all $j \in [p]$, and for every $\tilde{\mathbf{z}}$ with $\|\tilde{\mathbf{z}}\| \leq 2n$ the inputs to the neurons $\mathcal{E}_1, \mathcal{E}_2, \mathcal{E}_3$ in the computation $\hat{N}(\tilde{\mathbf{z}})$ satisfy Properties (P1) through (P3). We first show that with probability at least $1 - \frac{1}{n}$ all examples $\tilde{\mathbf{z}}_1, \ldots, \tilde{\mathbf{z}}_{m(n)+n^3}$ satisfy $\|\tilde{\mathbf{z}}_i\| \leq 2n$. Hence, with probability at least $1 - \frac{2}{n}$, Properties (P1) through (P3) hold for the computations $\hat{N}(\tilde{\mathbf{z}}_i)$ for all $i \in [m(n) + n^3]$.

Note that $\|\tilde{\mathbf{z}}_i\|^2$ has the Chi-squared distribution. Since $\tilde{\mathbf{z}}_i$ is of dimension $n^2$, a concentration bound by Laurent and Massart [32, Lemma 1] implies that for all $t > 0$ we have

$$\Pr\left[ \|\tilde{\mathbf{z}}_i\|^2 - n^2 \geq 2n\sqrt{t} + 2t \right] \leq e^{-t} \ .$$

Plugging-in $t = \frac{n^2}{4}$, we get

$$\Pr\left[ \|\tilde{\mathbf{z}}_i\|^2 \geq 4n^2 \right] = \Pr\left[ \|\tilde{\mathbf{z}}_i\|^2 - n^2 \geq 3n^2 \right]$$

$$\leq \Pr\left[ \|\tilde{\mathbf{z}}_i\|^2 - n^2 \geq \frac{3n^2}{2} \right]$$

$$= \Pr\left[ \|\tilde{\mathbf{z}}_i\|^2 - n^2 \geq 2n\sqrt{\frac{n^2}{4}} + 2 \cdot \frac{n^2}{4} \right]$$

$$\leq \exp\left( -\frac{n^2}{4} \right) \ .$$

Thus, we have $\Pr\left[ \|\tilde{\mathbf{z}}_i\| \geq 2n \right] \leq \exp\left( -\frac{n^2}{4} \right)$. By the union bound, with probability at least

$$1 - \left( m(n) + n^3 \right) \exp\left( -\frac{n^2}{4} \right) \geq 1 - \frac{1}{n}$$

(for a sufficiently large $n$), all examples $(\tilde{\mathbf{z}}_i, \tilde{y}_i)$ satisfy $\|\tilde{\mathbf{z}}_i\| \leq 2n$.

Thus, we showed that with probability at least $1 - \frac{2}{n} \geq \frac{39}{40}$ (for a sufficiently large $n$), we have $|\xi_j| \leq \frac{1}{10}$ for all $j \in [p]$, and Properties (P1) through (P3) hold for the computations $\hat{N}(\tilde{\mathbf{z}}_i)$ for all $i \in [m(n) + n^3]$. It remains to show that if these properties hold, then the examples $(\tilde{\mathbf{z}}_1, \tilde{y}_1), \ldots, (\tilde{\mathbf{z}}_{m(n)+n^3}, \tilde{y}_{m(n)+n^3})$ are realized by $\hat{N}$.

Let $i \in [m(n) + n^3]$. For brevity, we denote $\tilde{\mathbf{z}} = \tilde{\mathbf{z}}_i$, $\tilde{y} = \tilde{y}_i$, and $\mathbf{z}' = \tilde{\mathbf{z}}_{[kn]}$. Since $|\xi_j| \leq \frac{1}{10}$ for all $j \in [p]$, and all incoming weights to the output neuron in $\tilde{N}$ are $-1$, then in $\hat{N}$ all incoming weights to the output neuron are in $\left[-\frac{11}{10}, -\frac{9}{10}\right]$, and the bias term in the output neuron, denoted by $\hat{b}$, is in $\left[\frac{9}{10}, \frac{11}{10}\right]$. Consider the following cases:

- If $\Psi(\mathbf{z}')$ is not an encoding of a hyperedge then $\tilde{y} = 0$, and $\hat{N}(\tilde{\mathbf{z}})$ satisfies:

  1. If $\mathbf{z}'$ does not have components in $\left(c, c + \frac{1}{n}\right)$, then there exists a neuron in $\mathcal{E}_2$ with output at least $\frac{3}{2}$ (by Property (P2)).
  2. If $\mathbf{z}'$ has a component in $\left(c, c + \frac{1}{n}\right)$, then there exists a neuron in $\mathcal{E}_3$ with output at least $\frac{3}{2}$ (by Property (P3)).

  In both cases, since all incoming weights to the output neuron in $\hat{N}$ are in $\left[-\frac{11}{10}, -\frac{9}{10}\right]$, and $\hat{b} \in \left[\frac{9}{10}, \frac{11}{10}\right]$, then the input to the output neuron (including the bias term) is at most $\frac{11}{10} - \frac{3}{2} \cdot \frac{9}{10} < 0$, and thus its output is 0.

- If $\Psi(\mathbf{z}')$ is an encoding of a hyperedge $S$, then by the definition of the examples oracle we have $S = S_i$. Hence:

  - If $\mathbf{z}'$ does not have components in $\left(c - \frac{1}{n^2}, c + \frac{2}{n^2}\right)$, then:
    * If $y_i = 0$ then the oracle sets $\tilde{y} = \hat{b}$. Since $\mathcal{S}$ is pseudorandom, we have $P_{\mathbf{x}}(\mathbf{z}^S) = P_{\mathbf{x}}(\mathbf{z}^{S_i}) = y_i = 0$. Hence, in the computation $\hat{N}(\tilde{\mathbf{z}})$ the inputs to all neurons in $\mathcal{E}_1, \mathcal{E}_2, \mathcal{E}_3$ are at most $-\frac{1}{2}$ (by Properties (P1), (P2) and (P3)), and thus their outputs are 0. Therefore, $\hat{N}(\tilde{\mathbf{z}}) = \hat{b}$.
    * If $y_i = 1$ then the oracle sets $\tilde{y} = 0$. Since $\mathcal{S}$ is pseudorandom, we have $P_{\mathbf{x}}(\mathbf{z}^S) = P_{\mathbf{x}}(\mathbf{z}^{S_i}) = y_i = 1$. Hence, in the computation $\hat{N}(\tilde{\mathbf{z}})$ there exists a neuron in $\mathcal{E}_1$ with output at least $\frac{3}{2}$ (by Property (P1)). Since all incoming weights to the output neuron in $\hat{N}$ are in $\left[-\frac{11}{10}, -\frac{9}{10}\right]$, and $\hat{b} \in \left[\frac{9}{10}, \frac{11}{10}\right]$, then the input to output neuron (including the bias term) is at most $\frac{11}{10} - \frac{3}{2} \cdot \frac{9}{10} < 0$, and thus its output is 0.

  - If $\mathbf{z}'$ has a component in $\left(c, c + \frac{1}{n^2}\right)$, then $\tilde{y} = 0$. Also, in the computation $\hat{N}(\tilde{\mathbf{z}})$ there exists a neuron in $\mathcal{E}_3$ with output at least $\frac{3}{2}$ (by Property (P3)). Since all incoming weights to the output neuron in $\hat{N}$ are in $\left[-\frac{11}{10}, -\frac{9}{10}\right]$, and $\hat{b} \in \left[\frac{9}{10}, \frac{11}{10}\right]$, then the input to output neuron (including the bias term) is at most $\frac{11}{10} - \frac{3}{2} \cdot \frac{9}{10} < 0$, and thus its output is 0.

  - If $\mathbf{z}'$ does not have components in the interval $\left(c, c + \frac{1}{n^2}\right)$, but has a component in the interval $\left(c - \frac{1}{n^2}, c + \frac{2}{n^2}\right)$, then:
    * If $y_i = 1$ the oracle sets $\tilde{y} = 0$. Since $\mathcal{S}$ is pseudorandom, we have $P_{\mathbf{x}}(\mathbf{z}^S) = P_{\mathbf{x}}(\mathbf{z}^{S_i}) = y_i = 1$. Hence, in the computation $\hat{N}(\tilde{\mathbf{z}})$ there exists a neuron in $\mathcal{E}_1$ with output at least $\frac{3}{2}$ (by Property (P1)). Since all incoming weights to the output neuron in $\hat{N}$ are in $\left[-\frac{11}{10}, -\frac{9}{10}\right]$, and $\hat{b} \in \left[\frac{9}{10}, \frac{11}{10}\right]$, then the input to output neuron (including the bias term) is at most $\frac{11}{10} - \frac{3}{2} \cdot \frac{9}{10} < 0$, and thus its output is 0.
    * If $y_i = 0$ the oracle sets $\tilde{y} = [\hat{b} - \hat{N}_3(\tilde{\mathbf{z}})]_+$. Since $\mathcal{S}$ is pseudorandom, we have $P_{\mathbf{x}}(\mathbf{z}^S) = P_{\mathbf{x}}(\mathbf{z}^{S_i}) = y_i = 0$. Therefore, in the computation $\hat{N}(\tilde{\mathbf{z}})$ all neurons in $\mathcal{E}_1, \mathcal{E}_2$ have output 0 (by Properties (P1) and (P2)), and hence their contribution to the output of $\hat{N}$ is 0. Thus, by the definition of $\hat{N}_3$, we have $\hat{N}(\tilde{\mathbf{z}}) = [\hat{b} - \hat{N}_3(\tilde{\mathbf{z}})]_+$.

$\square$

**Lemma A.9.** *If $\mathcal{S}$ is pseudorandom, then for a sufficiently large $n$, with probability greater than $\frac{2}{3}$ we have*

$$\ell_I(h') \leq \frac{2}{n} \ .$$

*Proof.* By Lemma A.8, if $\mathcal{S}$ is pseudorandom then with probability at least $\frac{39}{40}$ (over $\boldsymbol{\xi} \sim \mathcal{N}(\mathbf{0}, \tau^2 I_p)$ and the i.i.d. inputs $\tilde{\mathbf{z}}_i \sim \mathcal{D}$) the examples $(\tilde{\mathbf{z}}_1, \tilde{y}_1), \ldots, (\tilde{\mathbf{z}}_{m(n)}, \tilde{y}_{m(n)})$ returned by the oracle are realized by $\hat{N}$. Recall that the algorithm $\mathcal{L}$ is such that with probability at least $\frac{3}{4}$ (over $\boldsymbol{\xi} \sim \mathcal{N}(\mathbf{0}, \tau^2 I_p)$, the i.i.d. inputs $\tilde{\mathbf{z}}_i \sim \mathcal{D}$, and possibly its internal randomness), given a size-$m(n)$ dataset labeled by $\hat{N}$, it returns a hypothesis $h$ such that $\mathbb{E}_{\tilde{\mathbf{z}} \sim \mathcal{D}}\left[(h(\tilde{\mathbf{z}}) - \hat{N}(\tilde{\mathbf{z}}))^2\right] \leq \frac{1}{n}$. Hence, with probability at least $\frac{3}{4} - \frac{1}{40}$ the algorithm $\mathcal{L}$ returns such a good hypothesis $h$, given $m(n)$ examples labeled by our examples oracle. Indeed, note that $\mathcal{L}$ can return a bad hypothesis only if the random choices are either bad for $\mathcal{L}$ (when used with realizable examples) or bad for the realizability of the examples returned by our oracle. By the definition of $h'$ and the construction of $\hat{N}$, if $h$ has small error then $h'$ also has small error, namely,

$$\mathbb{E}_{\tilde{\mathbf{z}} \sim \mathcal{D}}\left[(h'(\tilde{\mathbf{z}}) - \hat{N}(\tilde{\mathbf{z}}))^2\right] \leq \mathbb{E}_{\tilde{\mathbf{z}} \sim \mathcal{D}}\left[(h(\tilde{\mathbf{z}}) - \hat{N}(\tilde{\mathbf{z}}))^2\right] \leq \frac{1}{n} \ .$$

Let $\hat{\ell}_I(h') = \frac{1}{|I|} \sum_{i \in I} (h'(\tilde{\mathbf{z}}_i) - \hat{N}(\tilde{\mathbf{z}}_i))^2$. Recall that by our choice of $\tau$ we have $\Pr[\hat{b} > \frac{11}{10}] \leq \frac{1}{n}$. Since, $(h'(\tilde{\mathbf{z}}) - \hat{N}(\tilde{\mathbf{z}}))^2 \in [0, \hat{b}^2]$ for all $\tilde{\mathbf{z}} \in \mathbb{R}^{n^2}$, by Hoeffding's inequality, we have for a sufficiently large $n$ that

$$\begin{aligned}
\Pr\left[\left|\hat{\ell}_I(h') - \mathbb{E}_{\tilde{\mathcal{S}}_I}\hat{\ell}_I(h')\right| \geq \frac{1}{n}\right] &= \Pr\left[\left|\hat{\ell}_I(h') - \mathbb{E}_{\tilde{\mathcal{S}}_I}\hat{\ell}_I(h')\right| \geq \frac{1}{n}\,\bigg|\, \hat{b} \leq \frac{11}{10}\right] \cdot \Pr\left[\hat{b} \leq \frac{11}{10}\right] \\
&\quad + \Pr\left[\left|\hat{\ell}_I(h') - \mathbb{E}_{\tilde{\mathcal{S}}_I}\hat{\ell}_I(h')\right| \geq \frac{1}{n}\,\bigg|\, \hat{b} > \frac{11}{10}\right] \cdot \Pr\left[\hat{b} > \frac{11}{10}\right] \\
&\leq 2 \exp\left(-\frac{2n^3}{n^2(11/10)^4}\right) \cdot 1 + 1 \cdot \frac{1}{n} \\
&\leq \frac{1}{40} \ .
\end{aligned}$$

Moreover, by Lemma A.8,

$$\Pr\left[\ell_I(h') \neq \hat{\ell}_I(h')\right] \leq \Pr\left[\exists i \in I \text{ s.t. } \tilde{y}_i \neq \hat{N}(\tilde{\mathbf{z}}_i)\right] \leq \frac{1}{40} \ .$$

Overall, by the union bound we have with probability at least $1 - \left(\frac{1}{4} + \frac{1}{40} + \frac{1}{40} + \frac{1}{40}\right) > \frac{2}{3}$ for sufficiently large $n$ that:

- $\mathbb{E}_{\tilde{\mathcal{S}}_I}\hat{\ell}_I(h') = \mathbb{E}_{\tilde{\mathbf{z}} \sim \mathcal{D}}\left[(h'(\tilde{\mathbf{z}}) - \hat{N}(\tilde{\mathbf{z}}))^2\right] \leq \frac{1}{n}$.

- $\left|\hat{\ell}_I(h') - \mathbb{E}_{\tilde{\mathcal{S}}_I}\hat{\ell}_I(h')\right| \leq \frac{1}{n}$.

- $\ell_I(h') - \hat{\ell}_I(h') = 0$.

Combining the above, we get that if $\mathcal{S}$ is pseudorandom, then with probability greater than $\frac{2}{3}$ we have

$$\ell_I(h') = \left(\ell_I(h') - \hat{\ell}_I(h')\right) + \left(\hat{\ell}_I(h') - \mathbb{E}_{\tilde{\mathcal{S}}_I}\hat{\ell}_I(h')\right) + \mathbb{E}_{\tilde{\mathcal{S}}_I}\hat{\ell}_I(h') \leq 0 + \frac{1}{n} + \frac{1}{n} = \frac{2}{n} \ .$$

$\square$

**Lemma A.10.** *Let $\mathbf{z} \in \{0,1\}^{kn}$ be a random vector whose components are drawn i.i.d. from a Bernoulli distribution, which takes the value $0$ with probability $\frac{1}{n}$. Then, for a sufficiently large $n$, the vector $\mathbf{z}$ is an encoding of a hyperedge with probability at least $\frac{1}{\log(n)}$.*

*Proof.* The vector $\mathbf{z}$ represents a hyperedge iff in each of the $k$ size-$n$ slices in $\mathbf{z}$ there is exactly one 0-bit and each two of the $k$ slices in $\mathbf{z}$ encode different indices. Hence,

$$\Pr\left[\mathbf{z} \text{ represents a hyperedge}\right] = n \cdot (n-1) \cdot \ldots \cdot (n-k+1) \cdot \left(\frac{1}{n}\right)^k \left(\frac{n-1}{n}\right)^{nk-k}$$

$$\geq \left(\frac{n-k}{n}\right)^k \left(\frac{n-1}{n}\right)^{k(n-1)}$$

$$= \left(1 - \frac{k}{n}\right)^k \left(1 - \frac{1}{n}\right)^{k(n-1)} .$$

Since for every $x \in (0,1)$ we have $e^{-x} < 1 - \frac{x}{2}$, then for a sufficiently large $n$ the above is at least

$$\exp\left(-\frac{2k^2}{n}\right) \cdot \exp\left(-\frac{2k(n-1)}{n}\right) \geq \exp\left(-1\right) \cdot \exp\left(-2k\right) \geq \frac{1}{\log(n)} .$$

$\square$

**Lemma A.11.** *Let $\tilde{\mathbf{z}} \in \mathbb{R}^{n^2}$ be the vector returned by the oracle. We have*

$$\Pr\left[\tilde{\mathbf{z}} \in \tilde{\mathcal{Z}}\right] \geq \frac{1}{2\log(n)} .$$

*Proof.* Let $\mathbf{z}' = \tilde{\mathbf{z}}_{[kn]}$. We have

$$\Pr\left[\tilde{\mathbf{z}} \notin \tilde{\mathcal{Z}}\right] \leq \Pr\left[\exists j \in [kn] \text{ s.t. } z'_j \in \left(c - \frac{1}{n^2}, c + \frac{2}{n^2}\right)\right]$$

$$+ \Pr\left[\Psi(\mathbf{z}') \text{ does not represent a hyperedge}\right] . \quad (1)$$

We now bound the terms in the above RHS. First, since $\mathbf{z}'$ has the Gaussian distribution, then its components are drawn i.i.d. from a density function bounded by $\frac{1}{2\pi}$. Hence, for a sufficiently large $n$ we have

$$\Pr\left[\exists j \in [kn] \text{ s.t. } z'_j \in \left(c - \frac{1}{n^2}, c + \frac{2}{n^2}\right)\right] \leq kn \cdot \frac{1}{2\pi} \cdot \frac{3}{n^2} = \frac{3k}{2\pi n} \leq \frac{\log(n)}{n} . \quad (2)$$

Let $\mathbf{z} = \Psi(\mathbf{z}')$. Note that $\mathbf{z}$ is a random vector whose components are drawn i.i.d. from a Bernoulli distribution, where the probability to get 0 is $\frac{1}{n}$. By Lemma A.10, $\mathbf{z}$ is an encoding of a hyperedge with probability at least $\frac{1}{\log(n)}$. Combining it with Eq. (1) and (2), , we get for a sufficiently large $n$ that

$$\Pr\left[\tilde{\mathbf{z}} \notin \tilde{\mathcal{Z}}\right] \leq \frac{\log(n)}{n} + \left(1 - \frac{1}{\log(n)}\right) \leq 1 - \frac{1}{2\log(n)} ,$$

as required. $\square$

**Lemma A.12.** *If $\mathcal{S}$ is random, then for a sufficiently large $n$ with probability larger than $\frac{2}{3}$ we have*

$$\ell_I(h') > \frac{2}{n} .$$

*Proof.* Let $\tilde{\mathcal{Z}} \subseteq \mathbb{R}^{n^2}$ be such that $\tilde{\mathbf{z}} \in \tilde{\mathcal{Z}}$ iff $\tilde{\mathbf{z}}_{[kn]}$ does not have components in the interval $(c - \frac{1}{n^2}, c + \frac{2}{n^2})$, and $\Psi(\tilde{\mathbf{z}}_{[kn]}) = \mathbf{z}^S$ for a hyperedge $S$. If $\mathcal{S}$ is random, then by the definition of our examples oracle, for every $i \in [m(n) + n^3]$ such that $\tilde{\mathbf{z}}_i \in \tilde{\mathcal{Z}}$, we have $\tilde{y}_i = \hat{b}$ with probability $\frac{1}{2}$ and $\tilde{y}_i = 0$ otherwise. Also, by the definition of the oracle, $\tilde{y}_i$ is independent of $S_i$ and independent of the choice of the vector $\tilde{\mathbf{z}}_i$ that corresponds to $\mathbf{z}^{S_i}$. If $\hat{b} \geq \frac{9}{10}$ then for a sufficiently large $n$ the

hypothesis $h'$ satisfies for each random example $(\tilde{\mathbf{z}}_i, \tilde{y}_i) \in \tilde{\mathcal{S}}_I$ the following

$$\Pr_{(\tilde{\mathbf{z}}_i, \tilde{y}_i)} \left[ (h'(\tilde{\mathbf{z}}_i) - \tilde{y}_i)^2 \geq \frac{1}{5} \right]$$

$$\geq \Pr_{(\tilde{\mathbf{z}}_i, \tilde{y}_i)} \left[ (h'(\tilde{\mathbf{z}}_i) - \tilde{y}_i)^2 \geq \frac{1}{5} \,\middle|\, \tilde{\mathbf{z}}_i \in \tilde{\mathcal{Z}} \right] \cdot \Pr_{\tilde{\mathbf{z}}_i} \left[ \tilde{\mathbf{z}}_i \in \tilde{\mathcal{Z}} \right]$$

$$\geq \Pr_{(\tilde{\mathbf{z}}_i, \tilde{y}_i)} \left[ (h'(\tilde{\mathbf{z}}_i) - \tilde{y}_i)^2 \geq \left( \frac{\hat{b}}{2} \right)^2 \,\middle|\, \tilde{\mathbf{z}}_i \in \tilde{\mathcal{Z}} \right] \cdot \Pr_{\tilde{\mathbf{z}}_i} \left[ \tilde{\mathbf{z}}_i \in \tilde{\mathcal{Z}} \right]$$

$$\geq \frac{1}{2} \cdot \Pr_{\tilde{\mathbf{z}}_i} \left[ \tilde{\mathbf{z}}_i \in \tilde{\mathcal{Z}} \right] \;.$$

In Lemma A.11, we show that $\Pr_{\tilde{\mathbf{z}}_i} \left[ \tilde{\mathbf{z}}_i \in \tilde{\mathcal{Z}} \right] \geq \frac{1}{2\log(n)}$. Hence,

$$\Pr_{(\tilde{\mathbf{z}}_i, \tilde{y}_i)} \left[ (h'(\tilde{\mathbf{z}}_i) - \tilde{y}_i)^2 \geq \frac{1}{5} \right] \geq \frac{1}{2} \cdot \frac{1}{2\log(n)} \geq \frac{1}{4\log(n)} \;.$$

Thus, if $\hat{b} \geq \frac{9}{10}$ then we have

$$\mathbb{E}_{\tilde{\mathcal{S}}_I} \left[ \ell_I(h') \right] \geq \frac{1}{5} \cdot \frac{1}{4\log(n)} = \frac{1}{20\log(n)} \;.$$

Therefore, for large $n$ we have

$$\Pr \left[ \mathbb{E}_{\tilde{\mathcal{S}}_I} \left[ \ell_I(h') \right] \geq \frac{1}{20\log(n)} \right] \geq 1 - \frac{1}{n} \geq \frac{7}{8} \;.$$

Since, $(h'(\tilde{\mathbf{z}}) - \tilde{y})^2 \in [0, \hat{b}^2]$ for all $\tilde{\mathbf{z}}, \tilde{y}$ returned by the examples oracle, and the examples $\tilde{\mathbf{z}}_i$ for $i \in I$ are i.i.d., then by Hoeffding's inequality, we have for a sufficiently large $n$ that

$$\Pr \left[ \left| \ell_I(h') - \mathbb{E}_{\tilde{\mathcal{S}}_I} \ell_I(h') \right| \geq \frac{1}{n} \right] = \Pr \left[ \left| \ell_I(h') - \mathbb{E}_{\tilde{\mathcal{S}}_I} \ell_I(h') \right| \geq \frac{1}{n} \,\middle|\, \hat{b} \leq \frac{11}{10} \right] \cdot \Pr \left[ \hat{b} \leq \frac{11}{10} \right]$$

$$+ \Pr \left[ \left| \ell_I(h') - \mathbb{E}_{\tilde{\mathcal{S}}_I} \ell_I(h') \right| \geq \frac{1}{n} \,\middle|\, \hat{b} > \frac{11}{10} \right] \cdot \Pr \left[ \hat{b} > \frac{11}{10} \right]$$

$$\leq 2 \exp \left( -\frac{2n^3}{n^2 (11/10)^4} \right) \cdot 1 + 1 \cdot \frac{1}{n}$$

$$\leq \frac{1}{8} \;.$$

Hence, for large enough $n$, with probability at least $1 - \frac{1}{8} - \frac{1}{8} = \frac{3}{4} > \frac{2}{3}$ we have both $\mathbb{E}_{\tilde{\mathcal{S}}_I} \left[ \ell_I(h') \right] \geq \frac{1}{20\log(n)}$ and $\left| \ell_I(h') - \mathbb{E}_{\tilde{\mathcal{S}}_I} \ell_I(h') \right| \leq \frac{1}{n}$, and thus

$$\ell_I(h') \geq \frac{1}{20\log(n)} - \frac{1}{n} > \frac{2}{n} \;.$$

$\square$

## B  Proof of Corollary 3.1

By the proof of Theorem 3.1, under Assumption 2.1, there is no $\mathrm{poly}(d)$-time algorithm $\mathcal{L}_s$ that satisfies the following: Let $\boldsymbol{\theta} \in \mathbb{R}^p$ be $B$-bounded parameters of a depth-3 network $N_{\boldsymbol{\theta}} : \mathbb{R}^d \to \mathbb{R}$, and let $\tau, \epsilon > 0$. Assume that $p, B, 1/\epsilon, 1/\tau \leq \mathrm{poly}(d)$, and that the widths of the hidden layers in $\mathcal{N}_{\boldsymbol{\theta}}$ are $d$ (i.e., the weight matrices are square). Let $\boldsymbol{\xi} \in \mathcal{N}(\mathbf{0}, \tau^2 I_p)$ and let $\hat{\boldsymbol{\theta}} = \boldsymbol{\theta} + \boldsymbol{\xi}$. Then, with probability at least $\frac{3}{4} - \frac{1}{1000}$, given access to an examples oracle for $\mathcal{N}_{\hat{\boldsymbol{\theta}}}$, the algorithm $\mathcal{L}_s$ returns a hypothesis $h$ with $\mathbb{E}_{\mathbf{x}} \left[ (h(\mathbf{x}) - N_{\hat{\boldsymbol{\theta}}})^2 \right] \leq \epsilon$.

Note that in the above, the requirements from $\mathcal{L}_s$ are somewhat weaker than in our original definition of learning with smoothed parameters. Indeed, we assume that the widths of the hidden layers are $d$ and the required success probability is only $\frac{3}{4} - \frac{1}{1000}$ (rather than $\frac{3}{4}$). We now explain why the hardness result holds already under these conditions:

- Note that if we change the assumption on the learning algorithm in proof of Theorem 3.1 such that it succeeds with probability at least $\frac{3}{4} - \frac{1}{1000}$ (rather than $\frac{3}{4}$), then in the case where $\mathcal{S}$ is pseudorandom we get that the algorithm $\mathcal{A}$ returns 1 with probability at least $1 - \left(\frac{1}{4} + \frac{1}{1000} + \frac{1}{40} + \frac{1}{40} + \frac{1}{40}\right)$ (see the proof of Lemma A.9), which is still greater than $\frac{2}{3}$. Also, the analysis of the case where $\mathcal{S}$ is random does not change, and thus in this case $\mathcal{A}$ returns 0 with probability greater than $\frac{2}{3}$. Consequently, we still get distinguishing advantage greater than $\frac{1}{3}$.

- Regarding the requirement on the widths, we note that in the proof of Theorem 3.1 the layers satisfy the following. The input dimension is $d = n^2$, the width of the first hidden layer is at most $3n\log(n) \leq d$, and the width of the second hidden layer is at most $\log(n) + 2n + n\log(n) \leq d$ (all bounds are for a sufficiently large $d$). In order to get a network where all layers are of width $d$, we add new neurons to the hidden layers, with incoming weights 0, outgoing weights 0, and bias terms $-1$. Then, for an appropriate choice of $\tau = 1/\operatorname{poly}(n)$, even in the perturbed network the outputs of these new neurons will be 0 w.h.p. for every input $\tilde{\mathbf{z}}_1, \ldots, \tilde{\mathbf{z}}_{m(n)+n^3}$, and thus they will not affect the network's output. Thus, using the same argument as in the proof of Theorem 3.1, we conclude that the hardness results holds already for network with square weight matrices.

Suppose that there exists an efficient algorithm $\mathcal{L}_p$ that learns in the standard PAC framework depth-3 neural networks where the minimal singular value of each weight matrix is lower bounded by $1/q(d)$ for any polynomial $q(d)$. We will use $\mathcal{L}_p$ to obtain an efficient algorithm $\mathcal{L}_s$ that learns depth-3 networks with smoothed parameters as described above, and thus reach a contradiction.

Let $\boldsymbol{\theta} \in \mathbb{R}^p$ be $B$-bounded parameters of a depth-3 network $N_{\boldsymbol{\theta}} : \mathbb{R}^d \to \mathbb{R}$, and let $\tau, \epsilon > 0$. Assume that $p, B, 1/\epsilon, 1/\tau \leq \operatorname{poly}(d)$, and that the widths of the hidden layers in $\mathcal{N}_{\boldsymbol{\theta}}$ are $d$. For random $\boldsymbol{\xi} \sim \mathcal{N}(\mathbf{0}, \tau^2 I_p)$ and $\hat{\boldsymbol{\theta}} = \boldsymbol{\theta} + \boldsymbol{\xi}$, the algorithm $\mathcal{L}_s$ has access to examples labeled by $N_{\hat{\boldsymbol{\theta}}}$. Using Lemma B.1 below with $t = \frac{\tau}{d}$ and the union bound over the two weight matrices in $N_{\boldsymbol{\theta}}$, we get that with probability at least $1 - \frac{2 \cdot 2.35}{\sqrt{d}} \geq 1 - \frac{1}{1000}$ (for large enough $d$), the minimal singular values of all weight matrices in $\hat{\boldsymbol{\theta}}$ are at least $\frac{\tau}{d} \geq \frac{1}{q(d)}$ for some sufficiently large polynomial $q(d)$. Our algorithm $\mathcal{L}_s$ will simply run $\mathcal{L}_p$. Given that the minimal singular values of the weight matrices are at least $\frac{1}{q(d)}$, the algorithm $\mathcal{L}_p$ runs in time $\operatorname{poly}(d)$ and returns with probability at least $\frac{3}{4}$ a hypothesis $h$ with $\mathbb{E}_{\mathbf{x}}\left[(h(\mathbf{x}) - N_{\hat{\boldsymbol{\theta}}}(\mathbf{x}))^2\right] \leq \epsilon$. Overall, the algorithm $\mathcal{L}_s$ runs in $\operatorname{poly}(d)$ time, and with probability at least $\frac{3}{4} - \frac{1}{1000}$ (over both $\boldsymbol{\xi}$ and the internal randomness) returns a hypothesis $h$ with loss at most $\epsilon$.

**Lemma B.1** (Sankar et al. [37], Theorem 3.3). *Let $W$ be an arbitrary square matrix in $\mathbb{R}^{d \times d}$, and let $P \in \mathbb{R}^{d \times d}$ be a random matrix, where each entry is drawn i.i.d. from $\mathcal{N}(0, \tau^2)$ for some $\tau > 0$. Let $\sigma_d$ be the minimal singular value of the matrix $W + P$. Then, for every $t > 0$ we have*

$$\Pr_P\left[\sigma_d \leq t\right] \leq 2.35 \cdot \frac{t\sqrt{d}}{\tau} \ .$$

## C   Proof of Theorem 3.2

The proof follows similar ideas to the proof of Theorem 3.1. The main difference is that we need to handle here a smoothed discrete input distribution rather than the standard Gaussian distribution.

For a sufficiently large $n$, let $\mathcal{D}$ be a distribution on $\{0, 1\}^{n^2}$, where each component is drawn i.i.d. from a Bernoulli distribution which takes the value 0 with probability $\frac{1}{n}$. Assume that there is a $\operatorname{poly}(n)$-time algorithm $\mathcal{L}$ that learns depth-3 neural networks with at most $n^2$ hidden neurons and parameter magnitudes bounded by $n^3$, with smoothed parameters and inputs, under the distribution $\mathcal{D}$, with $\epsilon = \frac{1}{n}$ and $\tau, \omega = 1/\operatorname{poly}(n)$ that we will specify later. Let $m(n) \leq \operatorname{poly}(n)$ be the sample complexity of $\mathcal{L}$, namely, $\mathcal{L}$ uses a sample of size at most $m(n)$ and returns with probability at least $\frac{3}{4}$ a hypothesis $h$ with $\mathbb{E}_{\mathbf{z} \sim \hat{\mathcal{D}}}\left[(h(\mathbf{z}) - N_{\hat{\boldsymbol{\theta}}}(\mathbf{z}))^2\right] \leq \epsilon = \frac{1}{n}$. Note that $\hat{\mathcal{D}}$ is the distribution $\mathcal{D}$ after smoothing with parameter $\omega$, and the vector $\hat{\boldsymbol{\theta}}$ is the parameters of the target network after smoothing with parameter $\tau$. Let $s > 1$ be a constant such that $n^s \geq m(n) + n^3$ for every sufficiently large $n$. By Assumption 2.1, there exists a constant $k$ and a predicate $P : \{0, 1\}^k \to \{0, 1\}$, such that $\mathcal{F}_{P,n,n^s}$

is $\frac{1}{3}$-PRG. We will show an efficient algorithm $\mathcal{A}$ with distinguishing advantage greater than $\frac{1}{3}$ and thus reach a contradiction.

Throughout this proof, we will use some notations from the proof of Theorem 3.1. We repeat it here for convenience. For a hyperedge $S = (i_1, \ldots, i_k)$ we denote by $\mathbf{z}^S \in \{0,1\}^{kn}$ the following encoding of $S$: the vector $\mathbf{z}^S$ is a concatenation of $k$ vectors in $\{0,1\}^n$, such that the $j$-th vector has $0$ in the $i_j$-th coordinate and $1$ elsewhere. Thus, $\mathbf{z}^S$ consists of $k$ size-$n$ slices, each encoding a member of $S$. For $\mathbf{z} \in \{0,1\}^{kn}$, $i \in [k]$ and $j \in [n]$, we denote $z_{i,j} = z_{(i-1)n+j}$. That is, $z_{i,j}$ is the $j$-th component in the $i$-th slice in $\mathbf{z}$. For $\mathbf{x} \in \{0,1\}^n$, let $P_{\mathbf{x}} : \{0,1\}^{kn} \to \{0,1\}$ be such that for every hyperedge $S$ we have $P_{\mathbf{x}}(\mathbf{z}^S) = P(\mathbf{x}_S)$. For $\tilde{\mathbf{z}} \in \mathbb{R}^{n^2}$ we denote $\tilde{\mathbf{z}}_{[kn]} = (\tilde{z}_1, \ldots, \tilde{z}_{kn})$, namely, the first $kn$ components of $\tilde{\mathbf{z}}$ (assuming $n^2 \geq kn$).

## C.1 Defining the target network for $\mathcal{L}$

Since our goal is to use the algorithm $\mathcal{L}$ for breaking PRGs, in this subsection we define a neural network $\tilde{N} : \mathbb{R}^{n^2} \to \mathbb{R}$ that we will later use as a target network for $\mathcal{L}$. The network $\tilde{N}$ contains the subnetworks $N_1, N_2$ that we define below.

Let $N_1$ be a depth-1 neural network (i.e., one layer, with activations in the output neurons) with input dimension $kn$, at most $\log(n)$ output neurons, and parameter magnitudes bounded by $n^3$ (all bounds are for a sufficiently large $n$), which satisfies the following. We denote the set of output neurons of $N_1$ by $\mathcal{E}_1$. Let $\mathbf{z}' \in \{0,1\}^{kn}$ be an input to $N_1$ such that $\mathbf{z}' = \mathbf{z}^S$ for some hyperedge $S$. Thus, even though $N_1$ takes inputs in $\mathbb{R}^{kn}$, we consider now its behavior for an input $\mathbf{z}'$ with discrete components in $\{0,1\}$. Fix some $\mathbf{x} \in \{0,1\}^n$. Then, for $S$ with $P_{\mathbf{x}}(\mathbf{z}^S) = 0$ the inputs to all output neurons $\mathcal{E}_1$ are at most $-1$, and for $S$ with $P_{\mathbf{x}}(\mathbf{z}^S) = 1$ there exists a neuron in $\mathcal{E}_1$ with input at least $2$. Recall that our definition of a neuron's input includes the addition of the bias term. The construction of the network $N_1$ is given in Lemma A.2. Note that the network $N_1$ depends on $\mathbf{x}$. Let $N_1' : \mathbb{R}^{kn} \to \mathbb{R}$ be a depth-2 neural network with no activation function in the output neuron, obtained from $N_1$ by summing the outputs from all neurons $\mathcal{E}_1$.

Let $N_2$ be a depth-1 neural network (i.e., one layer, with activations in the output neurons) with input dimension $kn$, at most $2n$ output neurons, and parameter magnitudes bounded by $n^3$ (for a sufficiently large $n$), which satisfies the following. We denote the set of output neurons of $N_2$ by $\mathcal{E}_2$. Let $\mathbf{z}' \in \{0,1\}^{kn}$ be an input to $N_2$ (note that it has components only in $\{0,1\}$) . If $\mathbf{z}'$ is an encoding of a hyperedge then the inputs to all output neurons $\mathcal{E}_2$ are at most $-1$, and otherwise there exists a neuron in $\mathcal{E}_2$ with input at least $2$. The construction of the network $N_2$ is given in Lemma A.4. Let $N_2' : \mathbb{R}^{kn} \to \mathbb{R}$ be a depth-2 neural network with no activation function in the output neuron, obtained from $N_2$ by summing the outputs from all neurons $\mathcal{E}_2$.

Let $N' : \mathbb{R}^{kn} \to \mathbb{R}$ be a depth-2 network obtained from $N_1', N_2'$ as follows. For $\mathbf{z}' \in \mathbb{R}^{kn}$ we have $N'(\mathbf{z}') = [1 - N_1'(\mathbf{z}') - N_2'(\mathbf{z}')]_+$. The network $N'$ has at most $n^2$ neurons, and parameter magnitudes bounded by $n^3$ (all bounds are for a sufficiently large $n$). Finally, let $\tilde{N} : \mathbb{R}^{n^2} \to \mathbb{R}$ be a depth-2 neural network such that $\tilde{N}(\tilde{\mathbf{z}}) = N'(\tilde{\mathbf{z}}_{[kn]})$.

## C.2 Defining the noise magnitudes $\tau, \omega$ and analyzing the perturbed network under perturbed inputs

In order to use the algorithm $\mathcal{L}$ w.r.t. some neural network with parameters $\boldsymbol{\theta}$ and a certain input distribution, we need to implement an examples oracle, such that the examples are drawn from a smoothed input distribution, and labeled according to a neural network with parameters $\boldsymbol{\theta} + \boldsymbol{\xi}$, where $\boldsymbol{\xi}$ is a random perturbation. Specifically, we use $\mathcal{L}$ with an examples oracle where the input distribution $\hat{\mathcal{D}}$ is obtained from $\mathcal{D}$ by smoothing, and the labels correspond to a network $\hat{N} : \mathbb{R}^{n^2} \to \mathbb{R}$ obtained from $\tilde{N}$ (w.r.t. an appropriate $\mathbf{x} \in \{0,1\}^n$ in the construction of $N_1$) by adding a small perturbation to the parameters. The smoothing magnitudes $\omega, \tau$ of the inputs and the network's parameters (respectively) are such that the following hold.

We first choose the parameter $\tau = 1/\operatorname{poly}(n)$ as follows. Let $f_{\boldsymbol{\theta}} : \mathbb{R}^{n^2} \to \mathbb{R}$ be any depth-2 neural network parameterized by $\boldsymbol{\theta} \in \mathbb{R}^r$ for some $r > 0$ with at most $n^2$ neurons, and parameter magnitudes bounded by $n^3$ (note that $r$ is polynomial in $n$). Then, $\tau$ is such that with probability at least $1 - \frac{1}{n}$ over $\boldsymbol{\xi} \sim \mathcal{N}(\mathbf{0}, \tau^2 I_r)$, we have $|\xi_i| \leq \frac{1}{10}$ for all $i \in [r]$, and the network $f_{\boldsymbol{\theta} + \boldsymbol{\xi}}$ is such that

for every input $\tilde{\mathbf{z}} \in \mathbb{R}^{n^2}$ with $\|\tilde{\mathbf{z}}\| \leq n$ and every neuron we have: Let $a, b$ be the inputs to the neuron in the computations $f_{\boldsymbol{\theta}}(\tilde{\mathbf{z}})$ and $f_{\boldsymbol{\theta}+\boldsymbol{\xi}}(\tilde{\mathbf{z}})$ (respectively), then $|a - b| \leq \frac{1}{4}$. Thus, $\tau$ is sufficiently small, such that w.h.p. adding i.i.d. noise $\mathcal{N}(0, \tau^2)$ to each parameter does not change the inputs to the neurons by more than $\frac{1}{4}$. Note that such an inverse-polynomial $\tau$ exists, since when the network size, parameter magnitudes, and input size are bounded by some $\mathrm{poly}(n)$, then the input to each neuron in $f_{\boldsymbol{\theta}}(\tilde{\mathbf{z}})$ is $\mathrm{poly}(n)$-Lipschitz as a function of $\boldsymbol{\theta}$, and thus it suffices to choose $\tau$ that implies with probability at least $1 - \frac{1}{n}$ that $\|\boldsymbol{\xi}\| \leq \frac{1}{q(n)}$ for a sufficiently large polynomial $q(n)$ (see Lemma A.7 for details).

Next, we choose the parameter $\omega = 1/\mathrm{poly}(n)$ as follows. Let $f_{\boldsymbol{\theta}} : \mathbb{R}^{n^2} \to \mathbb{R}$ be any depth-2 neural network parameterized by $\boldsymbol{\theta}$ with at most $n^2$ neurons, and parameter magnitudes bounded by $n^3 + \frac{1}{10}$. Then, $\omega$ is such that for every $\mathbf{z} \in \{0, 1\}^{n^2}$, with probability at least $1 - \exp(-n/2)$ over $\boldsymbol{\zeta} \sim \mathcal{N}(\mathbf{0}, \omega^2 I_{n^2})$ the following holds for every neuron in the $f_{\boldsymbol{\theta}}$: Let $a, b$ be the inputs to the neuron in the computations $f_{\boldsymbol{\theta}}(\mathbf{z})$ and $f_{\boldsymbol{\theta}}(\mathbf{z} + \boldsymbol{\zeta})$ (respectively), then $|a - b| \leq \frac{1}{4}$. Thus, $\omega$ is sufficiently small, such that w.h.p. adding noise $\mathcal{N}(\mathbf{0}, \omega^2 I_{n^2})$ to the input $\mathbf{z}$ does not change the inputs to the neurons by more than $\frac{1}{4}$. Note that such an inverse-polynomial $\omega$ exists, since when the network size and parameter magnitudes are bounded by some $\mathrm{poly}(n)$, then the input to each neuron in $f_{\boldsymbol{\theta}}(\mathbf{z})$ is $\mathrm{poly}(n)$-Lipschitz as a function of $\mathbf{z}$, and thus it suffices to choose $\omega$ that implies with probability at least $1 - \exp(-n/2)$ that $\|\boldsymbol{\zeta}\| \leq \frac{1}{q(n)}$ for a sufficiently large polynomial $q(n)$ (see Lemma A.7 for details).

Let $\tilde{\boldsymbol{\theta}} \in \mathbb{R}^p$ be the parameters of the network $\tilde{N}$. Recall that the parameters vector $\tilde{\boldsymbol{\theta}}$ is the concatenation of all weight matrices and bias terms. Let $\hat{\boldsymbol{\theta}} \in \mathbb{R}^p$ be the parameters of $\hat{N}$, namely, $\hat{\boldsymbol{\theta}} = \tilde{\boldsymbol{\theta}} + \boldsymbol{\xi}$ where $\boldsymbol{\xi} \sim \mathcal{N}(\mathbf{0}, \tau^2 I_p)$. By our choice of $\tau$ and the construction of the networks $N_1, N_2$, with probability at least $1 - \frac{1}{n}$ over $\boldsymbol{\xi}$, for every $\mathbf{z} \in \{0, 1\}^{n^2}$ the following holds: Let $\boldsymbol{\zeta} \sim \mathcal{N}(\mathbf{0}, \omega^2 I_{n^2})$ and let $\hat{\mathbf{z}} = \mathbf{z} + \boldsymbol{\zeta}$. Then with probability at least $1 - \exp(-n/2)$ over $\boldsymbol{\zeta}$ the differences between inputs to all neurons in the computations $\hat{N}(\hat{\mathbf{z}})$ and $\tilde{N}(\mathbf{z})$ are at most $\frac{1}{2}$. Indeed, w.h.p. for all $\mathbf{z} \in \{0, 1\}^{n^2}$ the computations $\tilde{N}(\mathbf{z})$ and $\hat{N}(\mathbf{z})$ are roughly similar (up to change of $1/4$ in the input to each neuron), and w.h.p. the computations $\hat{N}(\mathbf{z})$ and $\hat{N}(\hat{\mathbf{z}})$ are roughly similar (up to change of $1/4$ in the input to each neuron). Thus, with probability at least $1 - \frac{1}{n}$ over $\boldsymbol{\xi}$, the network $\hat{N}$ is such that for every $\mathbf{z} \in \{0, 1\}^{n^2}$, we have with probability at least $1 - \exp(-n/2)$ over $\boldsymbol{\zeta}$ that the computation $\hat{N}(\hat{\mathbf{z}})$ satisfies the following properties, where $\mathbf{z}' := \mathbf{z}_{[kn]}$:

(Q1) If $\mathbf{z}' = \mathbf{z}^S$ for some hyperedge $S$, then the inputs to $\mathcal{E}_1$ satisfy:
- If $P_{\mathbf{x}}(\mathbf{z}^S) = 0$ the inputs to all neurons in $\mathcal{E}_1$ are at most $-\frac{1}{2}$.
- If $P_{\mathbf{x}}(\mathbf{z}^S) = 1$ there exists a neuron in $\mathcal{E}_1$ with input at least $\frac{3}{2}$.

(Q2) The inputs to $\mathcal{E}_2$ satisfy:
- If $\mathbf{z}'$ is an encoding of a hyperedge then the inputs to all neurons $\mathcal{E}_2$ are at most $-\frac{1}{2}$.
- Otherwise, there exists a neuron in $\mathcal{E}_2$ with input at least $\frac{3}{2}$.

### C.3 Stating the algorithm $\mathcal{A}$

Given a sequence $(S_1, y_1), \ldots, (S_{n^s}, y_{n^s})$, where $S_1, \ldots, S_{n^s}$ are i.i.d. random hyperedges, the algorithm $\mathcal{A}$ needs to distinguish whether $\mathbf{y} = (y_1, \ldots, y_{n^s})$ is random or that $\mathbf{y} = (P(\mathbf{x}_{S_1}), \ldots, P(\mathbf{x}_{S_{n^s}})) = (P_{\mathbf{x}}(\mathbf{z}^{S_1}), \ldots, P_{\mathbf{x}}(\mathbf{z}^{S_{n^s}}))$ for a random $\mathbf{x} \in \{0, 1\}^n$. Let $\mathcal{S} = ((\mathbf{z}^{S_1}, y_1), \ldots, (\mathbf{z}^{S_{n^s}}, y_{n^s}))$.

We use the efficient algorithm $\mathcal{L}$ in order to obtain distinguishing advantage greater than $\frac{1}{3}$ as follows. Let $\boldsymbol{\xi}$ be a random perturbation, and let $\hat{N}$ be the perturbed network as defined above, w.r.t. the unknown $\mathbf{x} \in \{0, 1\}^n$. Note that given a perturbation $\boldsymbol{\xi}$, only the weights in the second layer of the subnetwork $N_1$ in $\hat{N}$ are unknown, since all other parameters do not depend on $\mathbf{x}$. The algorithm $\mathcal{A}$ runs $\mathcal{L}$ with the following examples oracle. In the $i$-th call, the oracle first draws $\mathbf{z}' \in \{0, 1\}^{kn}$ such that each component is drawn i.i.d. from a Bernoulli distribution which takes the value $0$ with probability $\frac{1}{n}$. If $\mathbf{z}'$ is an encoding of a hyperedge then the oracle replaces $\mathbf{z}'$ with $\mathbf{z}^{S_i}$. Let $\mathbf{z} \in \{0, 1\}^{n^2}$ be such that $\mathbf{z}_{[kn]} = \mathbf{z}'$, and the other $n^2 - kn$ components of $\mathbf{z}$ are drawn i.i.d. from

a Bernoulli distribution which takes the value $0$ with probability $\frac{1}{n}$. Note that the vector $\mathbf{z}$ has the distribution $\mathcal{D}$, since replacing an encoding of a random hyperedge by an encoding of another random hyperedge does not change the distribution of $\mathbf{z}'$. Let $\hat{\mathbf{z}} = \mathbf{z} + \boldsymbol{\zeta}$, where $\boldsymbol{\zeta} \sim \mathcal{N}(\mathbf{0}, \omega^2 I_{n^2})$. Note that $\hat{\mathbf{z}}$ has the distribution $\hat{\mathcal{D}}$. Let $\hat{b} \in \mathbb{R}$ be the bias term of the output neuron of $\hat{N}$. The oracle returns $(\hat{\mathbf{z}}, \hat{y})$, where the labels $\hat{y}$ are chosen as follows:

- If $\mathbf{z}'$ is not an encoding of a hyperedge, then $\hat{y} = 0$.
- If $\mathbf{z}'$ is an encoding of a hyperedge:
    - If $y_i = 0$ we set $\hat{y} = \hat{b}$.
    - If $y_i = 1$ we set $\hat{y} = 0$.

Let $h$ be the hypothesis returned by $\mathcal{L}$. Recall that $\mathcal{L}$ uses at most $m(n)$ examples, and hence $\mathcal{S}$ contains at least $n^3$ examples that $\mathcal{L}$ cannot view. We denote the indices of these examples by $I = \{m(n) + 1, \ldots, m(n) + n^3\}$, and the examples by $\mathcal{S}_I = \{(\mathbf{z}^{S_i}, y_i)\}_{i \in I}$. By $n^3$ additional calls to the oracle, the algorithm $\mathcal{A}$ obtains the examples $\hat{\mathcal{S}}_I = \{(\hat{\mathbf{z}}_i, \hat{y}_i)\}_{i \in I}$ that correspond to $\mathcal{S}_I$. Let $h'$ be a hypothesis such that for all $\tilde{\mathbf{z}} \in \mathbb{R}^{n^2}$ we have $h'(\tilde{\mathbf{z}}) = \max\{0, \min\{\hat{b}, h(\tilde{\mathbf{z}})\}\}$, thus, for $\hat{b} \geq 0$ the hypothesis $h'$ is obtained from $h$ by clipping the output to the interval $[0, \hat{b}]$. Let $\ell_I(h') = \frac{1}{|I|} \sum_{i \in I} (h'(\hat{\mathbf{z}}_i) - \hat{y}_i)^2$. Now, if $\ell_I(h') \leq \frac{2}{n}$, then $\mathcal{A}$ returns 1, and otherwise it returns 0. We remark that the decision of our algorithm is based on $h'$ (rather than $h$) since we need the outputs to be bounded, in order to allow using Hoeffding's inequality in our analysis, which we discuss in the next subsection.

### C.4  Analyzing the algorithm $\mathcal{A}$

Note that the algorithm $\mathcal{A}$ runs in $\mathrm{poly}(n)$ time. We now show that if $\mathcal{S}$ is pseudorandom then $\mathcal{A}$ returns 1 with probability greater than $\frac{2}{3}$, and if $\mathcal{S}$ is random then $\mathcal{A}$ returns 1 with probability less than $\frac{1}{3}$. To that end, we use similar arguments to the proof of Theorem 3.1.

In Lemma C.1, we show that if $\mathcal{S}$ is pseudorandom then with probability at least $\frac{39}{40}$ (over $\boldsymbol{\xi} \sim \mathcal{N}(\mathbf{0}, \tau^2 I_p)$ and $\boldsymbol{\zeta}_i \sim \mathcal{N}(\mathbf{0}, \omega^2 I_{n^2})$ for all $i \in [m(n)]$) the examples $(\hat{\mathbf{z}}_1, \hat{y}_1), \ldots, (\hat{\mathbf{z}}_{m(n)}, \hat{y}_{m(n)})$ returned by the oracle are realized by $\hat{N}$. Recall that the algorithm $\mathcal{L}$ is such that with probability at least $\frac{3}{4}$ (over $\boldsymbol{\xi} \sim \mathcal{N}(\mathbf{0}, \tau^2 I_p)$, the i.i.d. inputs $\hat{\mathbf{z}}_i \sim \hat{\mathcal{D}}$, and possibly its internal randomness), given a size-$m(n)$ dataset labeled by $\hat{N}$, it returns a hypothesis $h$ such that $\mathbb{E}_{\hat{\mathbf{z}} \sim \hat{\mathcal{D}}}\left[(h(\hat{\mathbf{z}}) - \hat{N}(\hat{\mathbf{z}}))^2\right] \leq \frac{1}{n}$. Hence, with probability at least $\frac{3}{4} - \frac{1}{40}$ the algorithm $\mathcal{L}$ returns such a good hypothesis $h$, given $m(n)$ examples labeled by our examples oracle. Indeed, note that $\mathcal{L}$ can return a bad hypothesis only if the random choices are either bad for $\mathcal{L}$ (when used with realizable examples) or bad for the realizability of the examples returned by our oracle. By the definition of $h'$ and the construction of $\hat{N}$, if $h$ has small error then $h'$ also has small error, namely,

$$\mathbb{E}_{\hat{\mathbf{z}} \sim \hat{\mathcal{D}}}\left[(h'(\hat{\mathbf{z}}) - \hat{N}(\hat{\mathbf{z}}))^2\right] \leq \mathbb{E}_{\tilde{\mathbf{z}} \sim \hat{\mathcal{D}}}\left[(h(\hat{\mathbf{z}}) - \hat{N}(\hat{\mathbf{z}}))^2\right] \leq \frac{1}{n}.$$

Let $\hat{\ell}_I(h') = \frac{1}{|I|} \sum_{i \in I} (h'(\hat{\mathbf{z}}_i) - \hat{N}(\hat{\mathbf{z}}_i))^2$. Recall that by our choice of $\tau$ we have $\Pr[\hat{b} > \frac{11}{10}] \leq \frac{1}{n}$. Since, $(h'(\hat{\mathbf{z}}) - \hat{N}(\hat{\mathbf{z}}))^2 \in [0, \hat{b}^2]$ for all $\hat{\mathbf{z}} \in \mathbb{R}^{n^2}$, by Hoeffding's inequality, we have for a sufficiently large $n$ that

$$\Pr\left[\left|\hat{\ell}_I(h') - \mathbb{E}_{\hat{\mathcal{S}}_I} \hat{\ell}_I(h')\right| \geq \frac{1}{n}\right] = \Pr\left[\left|\hat{\ell}_I(h') - \mathbb{E}_{\hat{\mathcal{S}}_I} \hat{\ell}_I(h')\right| \geq \frac{1}{n} \,\middle|\, \hat{b} \leq \frac{11}{10}\right] \cdot \Pr\left[\hat{b} \leq \frac{11}{10}\right]$$

$$+ \Pr\left[\left|\hat{\ell}_I(h') - \mathbb{E}_{\hat{\mathcal{S}}_I} \hat{\ell}_I(h')\right| \geq \frac{1}{n} \,\middle|\, \hat{b} > \frac{11}{10}\right] \cdot \Pr\left[\hat{b} > \frac{11}{10}\right]$$

$$\leq 2 \exp\left(-\frac{2n^3}{n^2(11/10)^4}\right) \cdot 1 + 1 \cdot \frac{1}{n}$$

$$\leq \frac{1}{40}.$$

Moreover, by Lemma C.1,

$$\Pr\left[\ell_I(h') \neq \hat{\ell}_I(h')\right] \leq \Pr\left[\exists i \in I \text{ s.t. } \hat{y}_i \neq \hat{N}(\hat{\mathbf{z}}_i)\right] \leq \frac{1}{40} \ .$$

Overall, by the union bound we have with probability at least $1 - \left(\frac{1}{4} + \frac{1}{40} + \frac{1}{40} + \frac{1}{40}\right) > \frac{2}{3}$ for sufficiently large $n$ that:

- $\mathbb{E}_{\hat{\mathcal{S}}_I} \hat{\ell}_I(h') = \mathbb{E}_{\hat{\mathbf{z}}\sim\hat{\mathcal{D}}}\left[(h'(\hat{\mathbf{z}}) - \hat{N}(\hat{\mathbf{z}}))^2\right] \leq \frac{1}{n}$.

- $\left|\hat{\ell}_I(h') - \mathbb{E}_{\hat{\mathcal{S}}_I} \hat{\ell}_I(h')\right| \leq \frac{1}{n}$.

- $\ell_I(h') - \hat{\ell}_I(h') = 0$.

Combining the above, we get that if $\mathcal{S}$ is pseudorandom, then with probability greater than $\frac{2}{3}$ we have

$$\ell_I(h') = \left(\ell_I(h') - \hat{\ell}_I(h')\right) + \left(\hat{\ell}_I(h') - \mathbb{E}_{\hat{\mathcal{S}}_I} \hat{\ell}_I(h')\right) + \mathbb{E}_{\hat{\mathcal{S}}_I} \hat{\ell}_I(h') \leq 0 + \frac{1}{n} + \frac{1}{n} = \frac{2}{n} \ .$$

We now consider the case where $\mathcal{S}$ is random. For an example $\hat{\mathbf{z}}_i = \mathbf{z}_i + \boldsymbol{\zeta}_i$ returned by the oracle, we denote $\mathbf{z}'_i = (\mathbf{z}_i)_{[kn]} \in \{0,1\}^{kn}$. Thus, $\mathbf{z}'_i$ is the input that the oracle used before adding the $n^2 - kn$ additional components and adding noise $\boldsymbol{\zeta}_i$. Let $\mathcal{Z}' \subseteq \{0,1\}^{kn}$ be such that $\mathbf{z}' \in \mathcal{Z}'$ iff $\mathbf{z}' = \mathbf{z}^S$ for some hyperedge $S$. If $\mathcal{S}$ is random, then by the definition of our examples oracle, for every $i \in [m(n) + n^3]$ such that $\mathbf{z}'_i \in \mathcal{Z}'$, we have $\hat{y}_i = \hat{b}$ with probability $\frac{1}{2}$ and $\hat{y}_i = 0$ otherwise. Also, by the definition of the oracle, $\hat{y}_i$ is independent of $S_i$, independent of the $n^2 - kn$ additional components that where added, and independent of the noise $\boldsymbol{\zeta}_i \sim \mathcal{N}(\mathbf{0}, \omega^2 I_{n^2})$ that corresponds to $\hat{\mathbf{z}}_i$.

If $\hat{b} \geq \frac{9}{10}$ then for a sufficiently large $n$ the hypothesis $h'$ satisfies for each random example $(\hat{\mathbf{z}}_i, \hat{y}_i) \in \hat{\mathcal{S}}_I$ the following:

$$\Pr_{(\hat{\mathbf{z}}_i,\hat{y}_i)}\left[(h'(\hat{\mathbf{z}}_i) - \hat{y}_i)^2 \geq \frac{1}{5}\right]$$

$$\geq \Pr_{(\hat{\mathbf{z}}_i,\hat{y}_i)}\left[(h'(\hat{\mathbf{z}}_i) - \hat{y}_i)^2 \geq \frac{1}{5} \,\middle|\, \mathbf{z}'_i \in \mathcal{Z}'\right] \cdot \Pr\left[\mathbf{z}'_i \in \mathcal{Z}'\right]$$

$$\geq \Pr_{(\hat{\mathbf{z}}_i,\hat{y}_i)}\left[(h'(\hat{\mathbf{z}}_i) - \hat{y}_i)^2 \geq \left(\frac{\hat{b}}{2}\right)^2 \,\middle|\, \mathbf{z}'_i \in \mathcal{Z}'\right] \cdot \Pr\left[\mathbf{z}'_i \in \mathcal{Z}'\right]$$

$$\geq \frac{1}{2} \cdot \Pr\left[\mathbf{z}'_i \in \mathcal{Z}'\right] \ .$$

In Lemma A.10, we show that for a sufficiently large $n$ we have $\Pr\left[\mathbf{z}'_i \in \mathcal{Z}'\right] \geq \frac{1}{\log(n)}$. Hence,

$$\Pr_{(\hat{\mathbf{z}}_i,\hat{y}_i)}\left[(h'(\hat{\mathbf{z}}_i) - \hat{y}_i)^2 \geq \frac{1}{5}\right] \geq \frac{1}{2} \cdot \frac{1}{\log(n)} \geq \frac{1}{2\log(n)} \ .$$

Thus, if $\hat{b} \geq \frac{9}{10}$ then we have

$$\mathbb{E}_{\hat{\mathcal{S}}_I} [\ell_I(h')] \geq \frac{1}{5} \cdot \frac{1}{2\log(n)} = \frac{1}{10\log(n)} \ .$$

Therefore, for large $n$ we have

$$\Pr\left[\mathbb{E}_{\hat{\mathcal{S}}_I} [\ell_I(h')] \geq \frac{1}{10\log(n)}\right] \geq 1 - \frac{1}{n} \geq \frac{7}{8} \ .$$

Since, $(h'(\hat{\mathbf{z}}) - \hat{y})^2 \in [0, \hat{b}^2]$ for all $\hat{\mathbf{z}}, \hat{y}$ returned by the examples oracle, and the examples $\hat{\mathbf{z}}_i$ for $i \in I$ are i.i.d., then by Hoeffding's inequality, we have for a sufficiently large $n$ that

$$
\Pr\left[\left|\ell_I(h') - \mathop{\mathbb{E}}_{\hat{S}_I} \ell_I(h')\right| \geq \frac{1}{n}\right] = \Pr\left[\left|\ell_I(h') - \mathop{\mathbb{E}}_{\hat{S}_I} \ell_I(h')\right| \geq \frac{1}{n} \,\middle|\, \hat{b} \leq \frac{11}{10}\right] \cdot \Pr\left[\hat{b} \leq \frac{11}{10}\right]
$$
$$
+ \Pr\left[\left|\ell_I(h') - \mathop{\mathbb{E}}_{\hat{S}_I} \ell_I(h')\right| \geq \frac{1}{n} \,\middle|\, \hat{b} > \frac{11}{10}\right] \cdot \Pr\left[\hat{b} > \frac{11}{10}\right]
$$
$$
\leq 2 \exp\left(-\frac{2n^3}{n^2 (11/10)^4}\right) \cdot 1 + 1 \cdot \frac{1}{n}
$$
$$
\leq \frac{1}{8} \ .
$$

Hence, for large enough $n$, with probability at least $1 - \frac{1}{8} - \frac{1}{8} = \frac{3}{4} > \frac{2}{3}$ we have both $\mathbb{E}_{\hat{S}_I}[\ell_I(h')] \geq \frac{1}{10\log(n)}$ and $\left|\ell_I(h') - \mathbb{E}_{\hat{S}_I} \ell_I(h')\right| \leq \frac{1}{n}$, and thus

$$
\ell_I(h') \geq \frac{1}{10\log(n)} - \frac{1}{n} > \frac{2}{n} \ .
$$

Overall, if $S$ is pseudorandom then with probability greater than $\frac{2}{3}$ the algorithm $\mathcal{A}$ returns 1, and if $S$ is random then with probability greater than $\frac{2}{3}$ the algorithm $\mathcal{A}$ returns 0. Thus, the distinguishing advantage is greater than $\frac{1}{3}$. This concludes the proof of the theorem. It remains to prove the deffered lemma on the realizability of the examples returned by the examples oracle:

**Lemma C.1.** *If $S$ is pseudorandom then with probability at least $\frac{39}{40}$ over $\boldsymbol{\xi} \sim \mathcal{N}(\mathbf{0}, \tau^2 I_p)$ and $\boldsymbol{\zeta}_i \sim \mathcal{N}(\mathbf{0}, \omega^2 I_{n^2})$ for $i \in [m(n) + n^3]$, the examples $(\hat{\mathbf{z}}_1, \hat{y}_1), \ldots, (\hat{\mathbf{z}}_{m(n)+n^3}, \hat{y}_{m(n)+n^3})$ returned by the oracle are realized by $\hat{N}$.*

*Proof.* By our choice of $\tau$ and $\omega$ and the construction of $N_1, N_2$, with probability at least $1 - \frac{1}{n}$ over $\boldsymbol{\xi} \sim \mathcal{N}(\mathbf{0}, \tau^2 I_p)$, we have $|\xi_j| \leq \frac{1}{10}$ for all $j \in [p]$, and for every $\mathbf{z} \in \{0, 1\}^{n^2}$ the following holds: Let $\boldsymbol{\zeta} \sim \mathcal{N}(\mathbf{0}, \omega^2 I_{n^2})$ and let $\hat{\mathbf{z}} = \mathbf{z} + \boldsymbol{\zeta}$. Then with probability at least $1 - \exp(-n/2)$ over $\boldsymbol{\zeta}$ the inputs to the neurons $\mathcal{E}_1, \mathcal{E}_2$ in the computation $\hat{N}(\hat{\mathbf{z}})$ satisfy Properties (Q1) and (Q2). Hence, with probability at least $1 - \frac{1}{n} - (m(n) + n^3) \exp(-n/2) \geq 1 - \frac{2}{n}$ (for a sufficiently large $n$), $|\xi_j| \leq \frac{1}{10}$ for all $j \in [p]$, and Properties (Q1) and (Q2) hold for the computations $\hat{N}(\hat{\mathbf{z}}_i)$ for all $i \in [m(n) + n^3]$. It remains to show that if $|\xi_j| \leq \frac{1}{10}$ for all $j \in [p]$ and Properties (Q1) and (Q2) hold, then the examples $(\hat{\mathbf{z}}_1, \hat{y}_1), \ldots, (\hat{\mathbf{z}}_{m(n)+n^3}, \hat{y}_{m(n)+n^3})$ are realized by $\hat{N}$.

Let $i \in [m(n) + n^3]$. We denote $\hat{\mathbf{z}}_i = \mathbf{z}_i + \boldsymbol{\zeta}_i$, namely, the $i$-th example returned by the oracle was obtained by adding noise $\boldsymbol{\zeta}_i$ to $\mathbf{z}_i \in \{0, 1\}^{n^2}$. We also denote $\mathbf{z}'_i = (\mathbf{z}_i)_{[kn]} \in \{0, 1\}^{kn}$. Since $|\xi_j| \leq \frac{1}{10}$ for all $j \in [p]$, and all incoming weights to the output neuron in $\tilde{N}$ are $-1$, then in $\hat{N}$ all incoming weights to the output neuron are in $\left[-\frac{11}{10}, -\frac{9}{10}\right]$, and the bias term in the output neuron, denoted by $\hat{b}$, is in $\left[\frac{9}{10}, \frac{11}{10}\right]$. Consider the following cases:

- If $\mathbf{z}'_i$ is not an encoding of a hyperedge then $\hat{y}_i = 0$. Moreover, in the computation $\hat{N}(\hat{\mathbf{z}}_i)$, there exists a neuron in $\mathcal{E}_2$ with output at least $\frac{3}{2}$ (by Property (Q2)) . Since all incoming weights to the output neuron in $\hat{N}$ are in $\left[-\frac{11}{10}, -\frac{9}{10}\right]$, and $\hat{b} \in \left[\frac{9}{10}, \frac{11}{10}\right]$, then the input to the output neuron (including the bias term) is at most $\frac{11}{10} - \frac{3}{2} \cdot \frac{9}{10} < 0$, and thus its output is 0.

- If $\mathbf{z}'$ is an encoding of a hyperedge $S$, then by the definition of the examples oracle we have $S = S_i$. Hence:

  - If $y_i = 0$ then the oracle sets $\hat{y}_i = \hat{b}$. Since $S$ is pseudorandom, we have $P_{\mathbf{x}}(\mathbf{z}^S) = P_{\mathbf{x}}(\mathbf{z}^{S_i}) = y_i = 0$. Hence, in the computation $\hat{N}(\hat{\mathbf{z}}_i)$ the inputs to all neurons in $\mathcal{E}_1, \mathcal{E}_2$ are at most $-\frac{1}{2}$ (by Properties (Q1) and (Q2)), and thus their outputs are 0. Therefore, $\hat{N}(\hat{\mathbf{z}}_i) = \hat{b}$.

– If $y_i = 1$ then the oracle sets $\hat{y}_i = 0$. Since $\mathcal{S}$ is pseudorandom, we have $P_\mathbf{x}(\mathbf{z}^S) = P_\mathbf{x}(\mathbf{z}^{S_i}) = y_i = 1$. Hence, in the computation $\hat{N}(\hat{\mathbf{z}}_i)$ there exists a neuron in $\mathcal{E}_1$ with output at least $\frac{3}{2}$ (by Property (Q1)). Since all incoming weights to the output neuron in $\hat{N}$ are in $\left[-\frac{11}{10}, -\frac{9}{10}\right]$, and $\hat{b} \in \left[\frac{9}{10}, \frac{11}{10}\right]$, then the input to output neuron (including the bias term) is at most $\frac{11}{10} - \frac{3}{2} \cdot \frac{9}{10} < 0$, and thus its output is 0.

□

