# OpenReview forum: "Computational Complexity of Learning Neural Networks: Smoothness and Degeneracy"
_NeurIPS.cc/2023/Conference — NeurIPS 2023 poster_

### Official Review · Reviewer_fpWr · 2023-06-23

**Soundness:** 3 good
**Presentation:** 3 good
**Contribution:** 2 fair
**Rating:** 5
**Confidence:** 3

**Summary:**

The paper presents results on the nonexistence of efficient learning algorithms for $3$-depth ReLU networks with smoothed parameters and Gaussian input. It also proves that in general, there is no efficient learning algorithm for $2$-depth ReLU networks with smoothed parameters and smoothed inputs (the smoothness is applied to a specially constructed input, which is Bernoulli). The neural networks they analyze, unlike previous works, have a ReLU unit at the single output.

**Strengths:**

The paper is theoretical in nature and so it is good that it is rigorous in its analysis and seeks to provide enough detail to understand its proofs. The paper presents its results in a logical order.

**Weaknesses:**

I will first introduce my main concerns, and then other important observations to address. (I must say that I haven’t followed Section 4 with enough detail, though I do have some observations based on what I read.)

1) I am trying to understand the relevance of the learning problem studied by the paper. When talking about learning, from what I can recall, I have basically seen two approaches in the literature:

Case a) When training a neural network, e.g. using SGD, one of the things we are concerned about is the testing performance of the network after being trained for some number of iterations or number of epochs.

Case b) I have seen other works that are concerned on generalization issues in terms of how close is the empirical loss (with respect to samples of the input data) from the population loss (defined by the expected value with respect to the input data distribution), which such closeness depending on different parameters such as the number of samples being used, the width of the network, the depth of the network, etc.

Both of these cases tackle the issue of how well a neural network learns and generalizes. However, the paper seems to tackle the problem of actually learning a given neural network (see Definitions 2.1 and 2.2), where the true parameter of the neural network is chosen adversarially (and sometimes smoothed afterwards). In the paper, essentially, the learning is done by some algorithm that outputs some hypothesis that seeks to approximate the adversarial neural network. However, how is this relevant in the context of training neural networks and their testing/generalization performance? The type of works in Case a) and  Case b), though theoretical, are trying to understand some practical issues in neural network learning that is relevant to the general ML community, but I don’t see such immediate connection in this paper. In which situations do we care about “learning a neural network” instead of “training a neural network so it learns”? This needs more motivation and clarification.
Also, the hypothesis $h$ that is supposed to learn the neural network in Definitions 2.1 and 2.2 seems to belong to any arbitrary class of functions. Is this right? If so, why isn’t this intractable to compute? In my mind, it would make sense for $h$ to belong to the same class of neural networks as defined by $N_\theta$ --- maybe it is the case in Section 4, but I just don’t know; all I know is that in Section 2 and 3, this is not clear to me.
Curiously, thinking about Case a) above, the paragraph that starts at line 102 mentions stochastic gradient descent (SGD), which makes me think that previous works have addressed the question of learning in a more practically motivated way, since SGD is used in practice. Can the authors also comment on this? How does it relate to your paper?

2) The paper addresses computational complexity, and from reading the paper it seems that such computation is linked to the algorithm $\mathcal{L}$. The computation by $\mathcal{L}$ seems to be related to how many times the algorithm needs to access the oracle in order to compute the final hypothesis $h$. This number of times is, I believe, the sample complexity of the algorithm. So, are we talking about computational complexity or sample complexity in the paper? Is the paper concerned about sample complexity at all? This must be clear on the paper, probably since the beginning of it. As far as I can see, the concept of sample complexity was only referenced in line 251 of Section 4 without much more detail about its relevance. Please, address this, since sample complexity is very important in ML and learning in general.

3) The first part of Section 4 explains how the proof used by the authors is related to [15] and what is the additional challenge that the authors present, i.e., the handling of smoothing. However, besides the first paragraph of Section 4, there is no more indication of whether some of the constructions being used throughout the proof correspond to ones in [15] or not. It would be nice to know which parts of the proof were taken from previous papers that also study learning of neural networks, and which parts weren’t.

4) This comment applies to networks of $k$-depth where $k>2$. When considering the neural network in definition 2.1 and 2.2, as well as in the statement of Theorems 3.1, there is no mention on how the neurons are distributed in the feedforward network. Do all hidden layers have the same width? Are the number of neurons per width irrelevant in terms of learning? Since the paper is concerned with presenting negative results, I guess they consider specific constructions of the networks in order to prove their results. Is that correct? If so, could this be mentioned? In general, could there be some mention about the specific topology of the networks in terms of the widths per layer?

5) Naturally, the paper recognizes that it might be possible to obtain results of efficient learning for different assumptions and topologies, such as not including an activation function at the output. I believe the authors should investigate this case for $3$-depth networks because this will strengthen their paper’s contribution. For example, if efficient learning is demonstrated for $3$-depth networks without activation function at the output, this will be very interesting because it elucidates more the role of the extra activation in efficient learning! I know that theoretical works are hard to do because proofs can be very non-trivial; however, since there is at least one previous work showing the efficiency of learning $2$-depth networks [6], how difficult would it be to adapt their proof to the case of $3$-depth networks?

Other observations:

1) The paper mentions that the neural network architecture they focus on have a ReLU at the single output layer. Previous works seem to focus on linear output, i.e. regression. Is having a ReLU at the output layer closer to real world applications? What is the motivation for it?

2) Definition 2.1 and 2.2 says that $(x,y)$ is drawn iid; however, it seems that only $x$ must be drawn iid, since $y$ is a deterministic function of $x$. Please check.

3) I think a proof outline is needed in Section 4, probably right before subsection 4.1 to better understand how the rest of the proof is built. Though the subsection titles in section 4 indicate what is being done, how they are pieced together in order to better understand the overall proof is not clear to me.

Minor:

1) In the abstract, line 9 mentions the word “hard”. It would be better to instead use expressions in terms of efficiency of the computation, etc.

2) Line 35, it seems that it should say “bounded from below”.

3) In line 86 it says that considering standard Gaussian distributions is “perhaps the most natural choice of an input distribution”, why is that? In real world applications, practitioners don’t care about this type of input. It may be good to insert a better motivation for it.

**Questions:**

Please, see above, in the section "Weaknesses" of this review.

**Limitations:**

Yes.

---

> ### Author Rebuttal · Authors · 2023-08-09
>
> We thank the reviewer for their efforts.
>
> Regarding the weaknesses:
> 1. The paper studies PAC learning, as defined in Definitions 2.1, 2.2 and 2.3. Thus, we study whether there exists an efficient algorithm that returns w.h.p. a hypothesis with a small population loss. In this common notion of learning, the learning algorithm can be any efficient algorithm, and it is not restricted to standard algorithms such as SGD. Also, the returned hypothesis is not restricted. Therefore, our hardness results imply that (under Assumption 2.1) there is no efficient algorithm that returns w.h.p. a hypothesis with a small population loss. As a special case, it implies that SGD cannot learn neural networks efficiently (even under the smoothed analysis framework, Gaussian inputs, depth-3, etc.). In other words, the fact that our definitions do not have any restrictions on the algorithm and the returned hypothesis only makes our hardness results stronger. In this sense, our results are stronger than the hardness results that are specific to gradient methods, which are mentioned in the paragraph that starts at line 102.
> 2. Our results consider computational complexity. That is, we show (under Assumption 2.1) that there is no poly-time algorithm that learns the considered networks under the smoothed analysis framework. In the proof, we assume that such a poly-time learning algorithm exists, and therefore this algorithm must also have a polynomial sample complexity, and we use it to obtain a poly-time algorithm that breaks the PRG given a polynomial number of examples. Thus, we refer to the sample complexity of the algorithm in the proof for technical reasons, but the theorems proved in the paper consider computational complexity. Our results do not rule out the existence of learning algorithms with a polynomial sample complexity but super-polynomial time.
> 3. Indeed, in the first paragraph of Section 4 we compare our proof to [15]. While we build on the technique from [15], none of the parts in the proof is taken directly from [15], except for Lemma A.1 in the appendix.
> 4. The construction considers a specific architecture, but by a simple argument (which we use in the proof of Corollary 3.1 in Appendix B) we can see that the hardness results hold even if all layers are of the same width (e.g., when all layers are of width d, where d is the input dimension). We will add a remark about this issue in the camera-ready version.
> 5. Indeed, it is not clear whether learning depth-3 networks without activation in the output in the smoothed-analysis framework is hard. Despite efforts by the community, the existing algorithms for learning non-degenerate depth-2 networks were not extended to depth 3, and it is unknown whether such an extension is possible. Intuitively, we feel that obtaining such an extension will be challenging. We agree that it is an important direction for future research.
>
> Regarding the other observations:
> 1. In real-world applications, practitioners can choose whether to include activation in the output or not. We note that our hardness results can be easily shown to apply also in the case of depth-4 networks without activation in the output neuron (by adding a linear layer to the depth-3 network from our construction).
> 2. Sure, we can just draw $x$ i.i.d. and then $y$ is a deterministic function of $x$.
> 3. We will try to clarify this.

---

> > ### Comment · Reviewer_fpWr · 2023-08-14
> > **Reponse**
> >
> > Thanks to the reviewers for replying to my review. I still have some concerns.
> >
> > 1) The authors mentioned "In real-world applications, practitioners can choose whether to include activation in the output or not" as a response to the motivation of having a ReLU in the output layer. This is not convincing. I still believe the authors have not properly motivated the use of ReLU at the output layer, and I believe it is not well-motivated from a practical perspective either. This lack of motivation would make the problem studied in the paper less relevant. To the best of my knowledge, people use smooth activations in the output layer (including linear output). Here are some possible reasons why. Firstly, the output layer should output values in the range specified by the problem of interest -- i.e., whether it is classification or regression, the output must be in the range of the predictive values of interest. ReLU only outputs positive values, disregarding whatever negative value that comes into play. Secondly, ReLU is not a smooth function, and it may be possible that even small changes in the input are able to create abrupt changes in the output of the network, something not ideal, for example, in regression problems. The authors must find use cases for ReLU in the output layer, preferably in the literature, otherwise, I really don't see much relevance in studying such architectures. What is the motivation for the authors to focus on ReLU? It seems an arbitrary choice so far. There is no indication in the paper thus far.
> >
> > Now, I note that the authors have said "We note that our hardness results can be easily shown to apply also in the case of depth-4 networks without activation in the output neuron (by adding a linear layer to the depth-3 network from our construction)." Can the authors provide more explanation about this? How can we know this is actually doable? The authors wrote "by adding a linear layer to the depth-3 network from our construction", but, isn't the depth-3 network already supposed to have a ReLU layer at the end? What was the point of studying ReLU at the output layer then?
> > Also, networks with linear output layers have already been studied -- the authors mentioned the work [1], for example. Though it seems that [1] studies depth-3 networks, I wonder how much it takes to extend the work to depth-4. In any case, studying deeper networks with linear output layers that go beyond depth-4 seems to be an incremental contribution at best.
> >
> > 2) I now focus on the response number 4. From what I gather from the authors' response, it seems that in the proofs a specific network architecture is used, and that an extension is possible to neurons with equal width. Can more information be provided on this architecture? It is nowhere to be found in the main paper (unless I missed it). I find a little bit misleading that nothing is said about the specificity of the architecture in the main paper (other than having ReLU as activation functions and in its output layer, and the depth), which could make the reader believe that the results hold for any depth-3 neural network (or even depth-4 for linear outputs, as specified by the authors). I want to see how restrictive the architecture used in the proofs is in terms of the width of its neurons, the input dimension, etc. The authors mentioned that their methods can be extend to architectures of equal width, the width of the input dimension. How is this possible? Moreover, in practical applications, neural networks tend to be wide, and so often they have widths that are larger than the input dimension.
> >
> > I will update my score as our discussion develops.

---

> > > ### Author Response · Authors · 2023-08-14
> > >
> > > Regarding the activation in the output neuron and the choice of ReLU in general:
> > > - We used the ReLU activation because it is the most commonly used activation, and since all previous works on hardness of improper learning used this activation.
> > > - Some previous works on hardness of learning also included activations in the output neuron (see [14,15]).
> > > - The result easily extends to depth-4 networks without activation in the output. We will add a remark about it.
> > >
> > > Why does the result apply to depth-4 without activation in the output?
> > > - This follows from the same proof. The only difference is that in the reduction, the target network is the depth-4 network obtained from the depth-3 network in the current reduction, by connecting the current output neuron to the new output neuron with an edge of weight 1. The proof will still hold for the same arguments as in the current proof.
> > > - The point of having ReLU in the output is that it allows us to show hardness for depth-3 networks rather than depth-4.
> > > - I suppose that you mean [11] (rather than [1]). They indeed studied depth-3 networks without an activation in the output. However, they did not consider the smoothed-analysis framework, which is the focus of our paper. Their result does not imply hardness in the smoothed-analysis framework. Moreover, despite some efforts that we made in this direction, we do not see a way to extend their technique to this setting.
> > > - In general, showing a hardness result for depth $i$ is stronger than showing hardness for depth $j>i$.
> > >
> > > Regarding the architecture:
> > > - As we discuss in the preliminaries, we consider networks where all layers are fully-connected and have bias terms. The theorems shows that (under our assumption) learning such networks (in the smoothed-analysis framework) cannot be done in time polynomial in $d,p,B,1/\epsilon,1/\tau$. In order to show this, it suffices to show that learning is hard already under some specific widths of the layers. So for the correctness of our theorem, we do not really care about the widths.
> > > - The construction in our proof does not have equal widths, because as we discussed in the previous point we did not need it in Theorem 3.1. However, in the proof of Corollary 3.1 we needed equal widths (the width of the input dimension) and hence in this proof we explain how our construction from the proof of Theorem 3.1 can be easily extended to handle equal widths. The details about the widths of the construction in the proof of Theorem 3.1 is also summarized in the proof of Corollary 3.1 (although these technical details are not very interesting).
> > > - A similar extension to the one described in the proof of Corollary 3.1 can be done for widths larger than the input dimension.

---

> > > > ### Comment · Reviewer_fpWr · 2023-08-15
> > > > **Reponse**
> > > >
> > > > I am grateful to the authors for their prompt response.
> > > >
> > > > 1) The motivation for the ReLU in the output neuron is not satisfactory. Firstly, while it may be true that ReLU activation is the most commonly used activation (as the authors claim), it is not true that they are used in the output neuron (as I already said) --- only on hidden neurons to introduce the desired non-linearity. Thus, this argument adds no value to justifying a ReLU output neuron.
> > > > Secondly, what activation function is used in [14,15]? After a quick glance, it was hard to see which activation function [14] uses, and [15] seems to not use no activation function at all. Moreover, my concerns are not whether there is activation function or not per se, but the fact that a ReLU is being used, something without (much) relevance as I explained.
> > > > Adding a ReLU at the end of the network to prove hardness results seems to have a more marginal contribution when one realizes that there isn't much motivation for its use. I am not saying that the mathematics done in the paper are not worthy of praise -- what I believe is that the problem being studied does not lead to enough contribution worthy of a top-tier venue due to its lack of motivation.
> > > >
> > > > 2) Something is not clear to me. The way things are written in the paper (in both main theorems) and in the last response from the authors seem to claim that the hardness results from the paper (the fact that a neural network cannot be learned efficiently) is applied to **any** possible fully connected network (under the specified depths and the smoothness conditions if applicable). However, I disagree with this. What the main results of the paper seem to say instead is: "for the specific network that we used in the proof, we show that it is hard to learn it". Let me be more explicit with an example. Theorem 3.1 currently says: "Under Assumption 2.1, there is no efficient algorithm that learns depth-3 networks with smoothed parameters (in the sense of Definition 2.2) under the standard Gaussian distribution." Instead, what it should say, from what I can gather, is: "Under Assumption 2.1, there exists depth-3 networks with smoothed parameters (in the sense of Definition 2.2) under the standard Gaussian distribution for which no efficient algorithm can learn it."
> > > > How do I know that there isn't a specific architecture of neural networks of depth-3 for which there may exist an efficient algorithm to learn it? There is a gap here to jump from the case specified in the proof to the more general case. Maybe it is a matter of semantics, in which the contribution of the paper is affected. Maybe I am missing something. Can the authors clarify this?

---

> > > > > ### Author Response · Authors · 2023-08-15
> > > > >
> > > > > Regarding the activation in the output neuron:
> > > > > - In [15] there is ReLU activation in the output. See Theorem 11 there and note that in Section 2.4 of the preliminaries they say “ Unless stated otherwise, the output neuron also has a ReLU activation function”. (the theorem and section numbers are from their COLT version.)
> > > > > - In [14], all theorems are for networks of depth-2, where the output has the “clipping activation”, denoted $[\cdot]_{[0,1]}$. Namely, this activation clips the value to the interval $[0,1]$. They note in Section 2.3 (in their NeurIPS version) that this activation can be implemented using two ReLU neurons, at the cost of adding a new layer to the network.
> > > > > - As an additional example, consider: *Shamir, Ohad. "Distribution-specific hardness of learning neural networks." The Journal of Machine Learning Research 19.1 (2018): 1135-1163*. His hardness result from Theorem 5 is for depth-2 networks with a clipping activation $[\cdot]_{[0,1]}$ in the output neuron, similarly to [14].
> > > > > - If you still don’t like the concept of activation in the output, then, as I mentioned, you can think about the result as hardness of learning depth-4 networks under the smoothed-analysis framework. Prior to our work, it was unknown whether assuming Gaussian inputs and non-degeneracy (or smoothing) of the parameters may suffice for efficiently learning neural networks of arbitrary depth. Since it is known that these assumptions suffice for efficiently learning one-hidden-layer networks, it seemed plausible that they could also suffice for deep networks. We show that these assumptions do not suffice for neural networks in general, and that they do not suffice even for depth-3 networks with activation in the output or depth-4 without this activation. Thus, even if you think about our result as hardness for depth-4, the significance of the result is barely affected.
> > > > >
> > > > > Regarding the second issue:
> > > > > - Learnability is a property of a hypothesis class, not of a specific network. In this work, we consider the class of depth-3 networks with a bounded number of parameters (denoted by $p$) and bounded weights (denoted by $B$), and show that it cannot be learned efficiently (i.e., polynomial in these parameters and in the input dimension, and in $1/\epsilon$ and the smoothing parameters). This is a very standard definition of the hypothesis class, and hence our statement of the hardness result is completely standard. You may verify that this is the case in all prior works on hardness of learning neural networks.
> > > > > - As in any hardness result, in the reduction we construct a specific instance, namely, a specific network from this class, which has some specific properties.
> > > > > - Your suggested ways to phrase our result, namely *”for the specific network that we used in the proof, we show that it is hard to learn it”* and *“Under Assumption 2.1, there exists depth-3 networks with smoothed parameters (in the sense of Definition 2.2) under the standard Gaussian distribution for which no efficient algorithm can learn it”* are not well-defined, because learnability is not a property of a specific network. The statement must talk about a hypothesis class.
> > > > > - What you might suggest, is a statement of the form: “learning depth-$3$ networks is hard (under the smoothed analysis framework) even when all layers in the network have the same width”. This statement talks about the hypothesis class of balanced networks. But as I discussed in the previous comment, our result also holds for this class.

---

> > > > > > ### Comment · Reviewer_fpWr · 2023-08-15
> > > > > > **Response.**
> > > > > >
> > > > > > 1) Even though previous published works have used activation functions on the output of the network that are not used in practice and which lack relevance, I don't believe this is enough of a motivation for the current work.
> > > > > >
> > > > > > 2) Can you provide an insight on how much having a ReLU activation function at the output plays a role in the proving your hardness results?
> > > > > >
> > > > > > 3) It is good that your results can prove hardness of depth-4 networks under smoothness without activation at the output. Then, is it still an open question whether there is efficient learning or not for dept-3 neural networks when there is no ReLU at the output under smoothing?
> > > > > >
> > > > > > 4) I still don't find convincing how the learnability of networks is stated, but maybe it is just that I am not understanding something in the mathematical formulation. Your hypothesis class is not restricted, it can be anything apparently. So my question is, how do you know there **does not exist** a particular architecture of dept-3 with ReLU at the output that can be efficiently learned by the hypothesis class under smoothing?

---

> > > > > > > ### Author Response · Authors · 2023-08-15
> > > > > > >
> > > > > > > 1. So you can think about the current work as hardness for depth-4 networks.
> > > > > > > 2. In the proof, we construct a network that should correctly label data from our examples oracle. In order to construct this network we needed to include activation in the output neuron (there are, of course, many technical details which are required to understand our proof, which are discussed in the paper).
> > > > > > > 3. Yes. We discuss it in the paper in the "Discussion" section.
> > > > > > > 4. We consider the class of depth-3 fully-connected networks with $p$ parameters and $B$-bounded weights, and we want a learning algorithm that runs in time polynomial in $d,p,B$ (and in the other relevant parameters, as defined in Section 2). If such an efficient algorithm exists, then we show that we can break a PRG (in contradiction to Assumption 2.1). So the question that we study is whether there is an efficient algorithm for this class (under smoothed analysis and Gaussian inputs). Similar settings have been studied before in a huge number of works, on both lower- and upper-bounds (see our related work section). Of course, you can find a specific architecture that makes learning trivial. For example, you can consider an architecture where the widths of all layers are $1$. This is essentially like learning a single neuron and can be done efficiently. As we already mentioned before, we note that our hardness result holds already if you consider networks where the widths of all layers are equal to the input dimension $d$.

---

### Official Review · Reviewer_fAAD · 2023-07-04

**Soundness:** 3 good
**Presentation:** 2 fair
**Contribution:** 3 good
**Rating:** 6
**Confidence:** 2

**Summary:**

This paper studies the computational complexity of learning 3-layer neural networks under the standard Gaussian distribution. Specifically, under a standard cryptographic assumption on the existence of local pseudorandom generators, the authors show that there is no poly-time algorithm that can learn 3-layer networks under the smoothed analysis framework. As a corollary, they show learning 3-layer networks is hard even with assuming a lower bound on the smallest singular values of the weight matrices.

**Strengths:**

- It is an interesting question to understand the worst-case hardness of learning neural networks, and under what assumptions learning is possible. Prior work has shown 2-layer networks can be learned under the smoothed analysis framework. The current paper, however, shows that learning 3-layer networks in the smoothed setting is hard, which is quite an interesting result.
- At a high level the proof appears to be sound, however I admit that I do not work on complexity theory and am unable to verify the proofs fully.

**Weaknesses:**

The novelty of the paper seems limited, particularly in light of the related work [11]. [11] shows that learning three-layer networks is hard, under the Learning With Rounding (LWR) assumption. [11] considers three-layer networks with no activation in the last layer, while the current paper considers networks with an activation in the last layer, but this last-layer activation seems like a very minor difference. Can the authors please comment further on the novelty of the current paper in comparison to this prior work?

[11]  S. Chen, A. Gollakota, A. R. Klivans, and R. Meka. Hardness of noise-free learning for two-hidden-layer neural networks. arXiv preprint arXiv:2202.05258, 2022.

**Questions:**

- I find the main text of the paper, in particular section 4, to be quite difficult to understand. The proof of the main theorem is written in a lengthy paragraph format, and as such I think the paper could be improved by organizing the exposition in section 4. Some concrete suggestions are the following:
    - The authors could add a high-level proof sketch at the beginning of section 4 before diving into the details of the argument. This could be particularly helpful to readers who are not familiar with the argument of [15], and complexity theory more generally (and may make the paper more suitable for a NeurIPS audience).
    - Key lemmas (such as Lemmas A.4 - A.6) could be stated explicitly in the main text, to make more clear what the various steps in the proof are.
    - Both the examples oracle and algorithm $\mathcal{A}$ could be written in algorithm format, to make more clear what exactly they are computing.
- I am a bit confused about what the point of the examples oracle is in Section 4.3, and how this is being used by the algorithm. Could you please clarify this further?
- As mentioned in the weaknesses section, the paper would benefit from additional comparison to [11], in particular the novelty and a comparison between the LWR and PRG cryptographic assumptions.

[15] A. Daniely and G. Vardi. From local pseudorandom generators to hardness of learning. In Conference on Learning Theory, pages 1358–1394. PMLR, 2021.

**Limitations:**

Limitations are adequately addressed.

---

> ### Author Rebuttal · Authors · 2023-08-09
>
> We thank the reviewer for their comments and suggestions.
>
> Regarding the comparison to [11]:
> In [11], the authors showed hardness of learning depth-3 networks under the Gaussian input distribution, but the neural network in their construction is degenerate. Hence, their result does not imply hardness of learning non-degenerate networks, and does not imply hardness under the smoothed-analysis framework. Moreover, despite some efforts that we made in this direction, we do not see a way to extend their technique to these settings. The novelty in the current paper is that we establish hardness results for non-degenerate networks and hardness under the smoothed-analysis framework. We view this contribution as significant and surprising, as for depth-2 networks the non-degeneracy assumption makes the problem solvable in poly time. As the reviewer mentioned, there are additional distinctions between our results and [11], which we view as more minor, such as different cryptographic assumptions and the existence of activation in the output neuron. As we discuss in the paper, our cryptographic assumption is considered established, but we do not think that there is a formal way to make a comparison to their LWR assumption (it is essentially a matter of taste).
>
> Regarding the readability of the proof in Section 4: We will try to apply the reviewer's suggestions to make the proof a bit easier to follow.
>
> Regarding the examples oracle in Section 4.3: As we define in Definition 2.2, a learning algorithm has access to an examples oracle, and returns w.h.p. a hypothesis that performs well on fresh examples from the examples oracle. In our proof, we assume that we have an efficient learning algorithm for smoothed depth-3 neural networks, and we run it with the examples oracle defined in Section 4.3 in order to break the PRG as follows. If the hypothesis returned by the learning algorithm performs well on fresh examples from the examples oracle then the data is pseudorandom, and otherwise it is random. If the reviewer has further questions on this issue we will be happy to elaborate more.

---

> > ### Comment · Reviewer_fAAD · 2023-08-15
> > **Response to Authors**
> >
> > Thank you to the authors for answering my questions. The novelty of the submission in comparison to the prior work is now more clear to me, and as such I am inclined to increase my score.

---

### Official Review · Reviewer_nyVW · 2023-07-04

**Soundness:** 4 excellent
**Presentation:** 4 excellent
**Contribution:** 3 good
**Rating:** 5
**Confidence:** 3

**Summary:**

The submission studies the classical problem of constructing ReLU-activated neural networks, specifically from the learning point of view. Previous work has established the existence of an efficient learning learning algorithm for learning depth-2 ReLU networks under the Gaussian distribution assuming a non-degenerate weight matrix. The authors provide a lower bound that rules out an extension of this result to depth-3 ReLU networks even under the "smoothed analysis" framework, where one assumes the presence of random noise to avoid the use of "degenerate" instances in the hardness reduction. As a second result, the authors provide a lower bound that rules out the extension of the aforementioned algorithm on the Gaussian distribution to general smoothed distributions.

**Strengths:**

The construction of ReLU-activated neural networks is a central topic at NeurIPS, and other lower-bound results on the problem have also appeared at the proceedings. The article is well-written and the results are non-trivial.

**Weaknesses:**

My main issue with the submission is that the newly obtained lower bounds seem to only shift the boundaries of intractability by a very small step compared to what was known previously. In particular, previous results of Daniely and Vardi [15] already ruled out (under the same complexity-theoretic assumptions) an efficient learning algorithm for the setting studied in this submission, with the distinction being that their lower bound does not operate in the "smoothed analysis" framework. The same work of Daniely and Vardi [15] also established a lower bound that matches the second result in this submission, but once again without the use of the "smoothed analysis" framework. In this sense, it seems to me that the main contribution of the submission is showing that previously established lower bounds do not require the use of purely degenerate cases. In combination with the fact that the lower-bound proof itself builds on the approach introduced by Daniely and Vardi [15], I cannot help but feel that the present submission is somewhat incremental in nature.

Also, a clearer and more to-the-point comparison of the submission's results to those obtained in the previous work(s) of Daniely and Vardi would have been appreciated... but that is only a minor (and perhaps subjective) point.

**Questions:**

N/A; however, the authors are of course welcome to respond to the individual points raised in the review.

**Limitations:**

The limitations have been adequately addressed.

---

> ### Author Rebuttal · Authors · 2023-08-09
>
> We thank the reviewer for their efforts.
>
> Our hardness results show that learning neural networks under the Gaussian distribution is hard already for non-degenerate instances. This is in contrast to all previously known hardness results for learning neural networks. Since for depth-2 networks the non-degeneracy assumption makes the problem solvable in polynomial time, we think that our hardness result for depth-3 is surprising. Also, from the technical point of view, the required construction is highly non-trivial, and differs significantly from [15]. Thus, we believe that the contribution of the current paper is significant both at the conceptual and technical levels.

---

> > ### Comment · Reviewer_nyVW · 2023-08-14
> >
> > Thank you for your response; I have read it and will keep it in mind during the discussion.

---

### Official Review · Reviewer_Hkzt · 2023-07-08

**Soundness:** 4 excellent
**Presentation:** 4 excellent
**Contribution:** 4 excellent
**Rating:** 7
**Confidence:** 1

**Summary:**

  This paper addresses the complexity of learning neural networks, a
very fundamental problem in learning theory. Previously, some
complexity and efficient solvability results were known for networks
of depth 2.  The paper shows that when the depth is increased to
3, the learning problem becomes computationally intractable (under
a hypothesis on the existence of local pseudorandom generators)
even when one uses the assumptions (e.g., Gaussian input distribution,
non-degeneracy of the weight matrix) that lead to efficient algorithms
for the depth-2 case.  The paper also shows that the learning problem
for depth-2 networks is hard under a smoothed analysis framework
(where both the input distribution and the network parameters are
perturbed).

**Strengths:**

(a) Understanding the boundary between hard and easy cases of learning
    neural networks is an important problem in learning theory. This is a
    very nice contribution to that area.

   (b) The paper provides a nice summary of previous work which makes
   it easier to understand the context for the contributions.

   (c) The technical results are presented very well.

**Weaknesses:**

 This reviewer can't see any weaknesses.

**Questions:**

(1) You prove hardness results for depth-3 neural nets. Can the proofs
  be extended (in a simple way) to show that the hardness results also hold for
  depth-k neural nets for any k >= 3?

  (2) Assumption 2.1 (on the existence of local PRGs with certain properties)
  is used in your proofs. You cite references that provide evidence for the
  assumption. It will be useful to the readers if you can include a short discussion
  on the evidence in the paper itself. Are there any consequences if
  Assumption 2.1 does not hold?

---

> ### Author Rebuttal · Authors · 2023-08-09
>
> We thank the reviewer for the positive review.
>
> Regarding the questions:
> 1. Yes, the result can be easily extended to any $k \geq 3$ by appending to our construction additional layers. We will add a remark about it in the camera-ready version.
> 2. Indeed, we referred the reader to papers where this assumption is discussed. One notable evidence for our assumption was shown by Applebaum [1]. He showed that our assumption follows from a variant of Goldreich’s one-wayness assumption (Goldreich, 2000). In addition, there is a concrete candidate for a local PRG, which is based on the XOR-MAJ predicate, and was shown to be secure against all known attacks. A more detailed discussion of our assumption can be found in [15, Section 2.2]. We will add some details on this subject in the camera-ready version.

---

> > ### Comment · Reviewer_Hkzt · 2023-08-12
> >
> >    I have gone through the rebuttal. My questions/concerns have been addressed in a satisfactory
> > fashion.

---

### Decision · Program_Chairs · 2023-09-21

**Decision:**

Accept (poster)

**Comment:**

The paper studies the computational complexity of training neural networks. In particular, the authors show that learning depth-3 NNs (technically their network is depth-4) is hard even under the smoothed analysis setting compared to previous known lower bounds that depended on degenerate cases. They use a cleverer variant of the so-called Daniely-Vardi lift that is resistant to smoothing at the cost of an extra layer (technically two non-linear layers) which allows them to embed depth-$L$ hard boolean functions as depth-$(L+2)$ ReLU networks.

The main concern raised by most reviewers was about the technical significance of extending the known hardness results to the "smoothed analysis" setting. Reviewer fpWr had additional concerns about the motivation of studying computational complexity in the PAC learning setting. Since the work is built on a long line of prior work studying this setting, and the setting is very classically studied in learning theory, I do not think the latter concern is well-founded. Additionally, I think that the "smoothed analysis" setting is technically significant in light of the brittleness of existing lower bounds to even slight perturbations. Therefore, I vote to accept the paper. I encourage the authors to add more discussion about the importance of the setting and comparison to prior work, and also clarify depth-4 vs depth-3 in their notation.